# Allosteric communication in DNA polymerase clamp loaders relies on a critical hydrogen-bonded junction

**Subu Subramanian[1,2,3], Kent Gorday[1,2,4], Kendra Marcus[1,2], Matthew R Orellana[1,2†], Peter Ren[1,2‡], Xiao Ran Luo[1,2], Michael E O'Donnell[5], John Kuriyan[1,2,3,6,7]\***

[1]Department of Molecular and Cell Biology, University of California, Berkeley, Berkeley, United States; [2]California Institute for Quantitative Biosciences (QB3), University of California, Berkeley, Berkeley, United States; [3]Howard Hughes Medical Institute, University of California, Berkeley, Berkeley, United States; [4]Biophysics Graduate Group, University of California, Berkeley, Berkeley, United States; [5]Howard Hughes Medical Institute, Rockefeller University, New York, United States; [6]Department of Chemistry, University of California, Berkeley, Berkeley, United States; [7]Physical Biosciences Division, Lawrence Berkeley National Laboratory, Berkeley, United States

**\*For correspondence:**
jkuriyan@mac.com

**Present address:** [†]Catalent, Harmans, United Stataes; [‡]DE Shaw Research, New York, United States

**Abstract** Clamp loaders are AAA+ ATPases that load sliding clamps onto DNA. We mapped the mutational sensitivity of the T4 bacteriophage sliding clamp and clamp loader by deep mutagenesis, and found that residues not involved in catalysis or binding display remarkable tolerance to mutation. An exception is a glutamine residue in the AAA+ module (Gln 118) that is not located at a catalytic or interfacial site. Gln 118 forms a hydrogen-bonded junction in a helical unit that we term the central coupler, because it connects the catalytic centers to DNA and the sliding clamp. A suppressor mutation indicates that hydrogen bonding in the junction is important, and molecular dynamics simulations reveal that it maintains rigidity in the central coupler. The glutamine-mediated junction is preserved in diverse AAA+ ATPases, suggesting that a connected network of hydrogen bonds that links ATP molecules is an essential aspect of allosteric communication in these proteins.

## Introduction

Sliding DNA clamps and the ATP-driven clamp-loader complexes that load them onto DNA are essential components of the DNA replication machinery in all branches of life (*Yao and O'Donnell, 2016*). Sliding clamps, such as proliferating cell nuclear antigen (PCNA) in eukaryotes, are ring-shaped proteins that enable highly processive DNA replication by tethering DNA polymerases to the template, allowing thousands of nucleotides to be incorporated into the growing strand without dissociation of the polymerase (*Hedglin et al., 2013*; *Kelch, 2016*; *Kelch et al., 2012*; *Oakley, 2019*; *Figure 1*). In addition to their essential role in replication, sliding clamps provide mobile platforms on DNA for proteins involved in DNA repair and chromatin remodeling (*Janke et al., 2018*; *Moldovan et al., 2007*). Sliding clamps form closed rings, and so they cannot bind to DNA until they are opened and loaded onto primer-template junctions by clamp-loader complexes.

All clamp-loader complexes consist of five subunits that have arisen through gene duplication of an ancestral ATPase subunit. The ATPase subunits of clamp loaders are members of the very large

**Figure 1.** Clamp-loader complex of T4 bacteriophage. (**A**) Crystal structure (left) and schematic diagram (right) of the clamp loader. (**B**) The clamp loading cycle, from left to right, showing the key stages of loading the sliding clamp around primer-templated DNA. (**C**) Key elements of AAA+ modules, shown here at the interface between neighboring ATPase subunits at positions B and C with DNA and the sliding clamp.

and diverse family of AAA+ ATPases, which are oligomeric proteins that bind and hydrolyze ATP at interfacial sites (*Gates and Martin, 2020*; *Neuwald et al., 1999*). Each subunit in a clamp-loader complex contains a two-domain AAA+ module that is characteristic of the AAA+ ATPases. The AAA+ module in each subunit is connected to a C-terminal collar domain that is responsible for olig-omerization of the clamp loader (*Bowman et al., 2004*; *Guenther et al., 1997*; *Jeruzalmi et al., 2001*; *Figure 1A*). The five subunits of the clamp loader are referred to as the A, B, C, D, and E sub-units, and the AAA+ modules within a complex form a spiral assembly when bound to ATP (*Bowman et al., 2004*). In some clamp loaders, the subunits at the A or E positions are degenerate and have lost the capacity to bind or hydrolyze ATP.

The clamp-loader complex is referred to as a 'molecular matchmaker' because each productive cycle of ATP hydrolysis brings together a sliding clamp and DNA, which do not otherwise form a sta-ble interaction (*Sancar and Hearst, 1993*). The clamp-loader cycle consists of three steps (*Figure 1B*). First, the ATP-bound clamp loader binds to and opens the sliding clamp (*Douma et al.,*

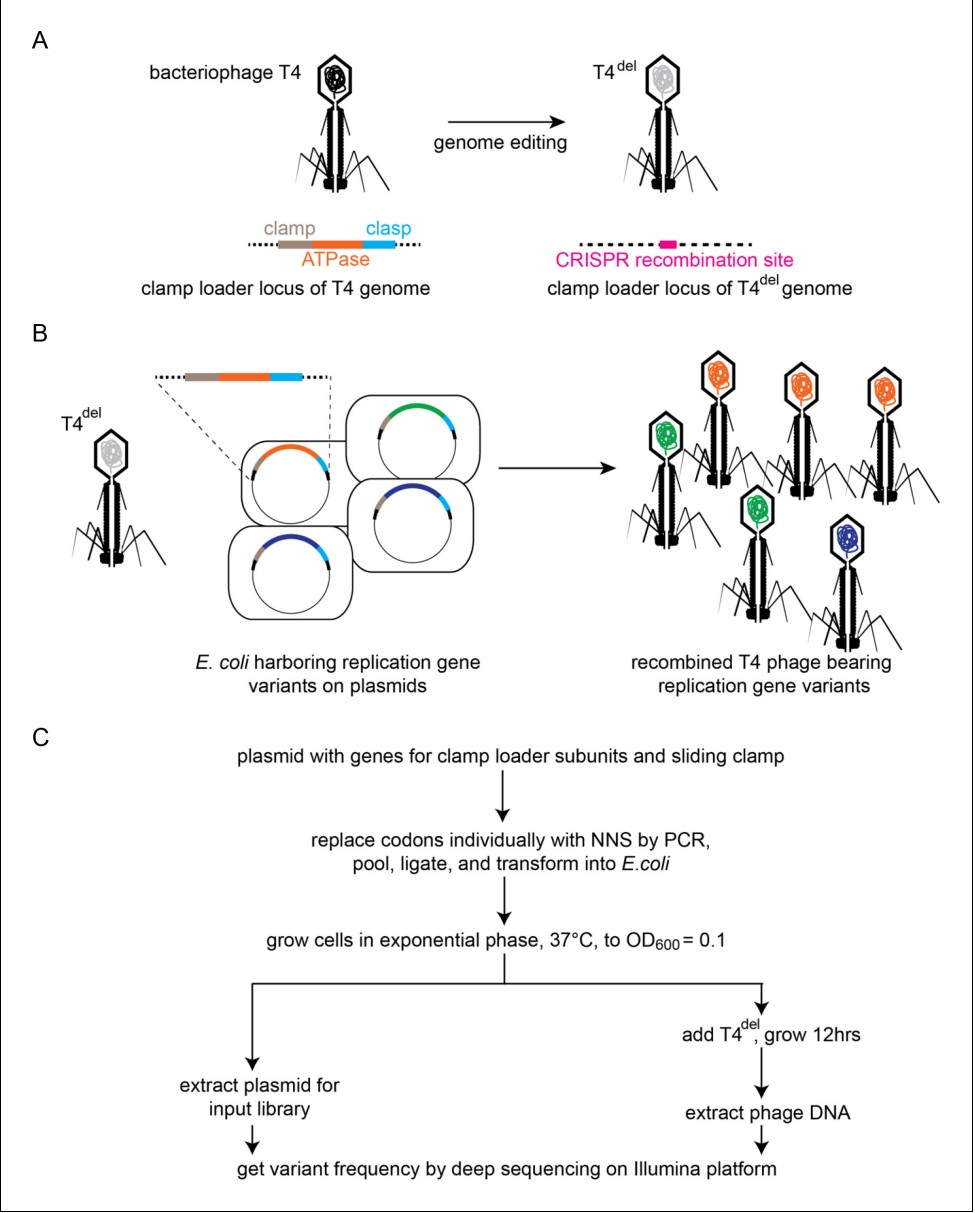

**Figure 2.** Clamp-loader activity assay through phage propagation. (**A**) Schematic depicts the generation of T4$^{del}$. The clamp-loader locus in wild-type T4 bacteriophage, containing genes for the sliding clamp and the ATPase and clasp subunits of the clamp loader, is replaced with a CRISPR-cas12 target site. See methods for details. (**B**) In the high-throughput phage-propagation assay, T4$^{del}$ infects bacteria carrying (i) a plasmid-encoded CRISPR-cas12 (not shown) programmed to target the recombination site inserted in A and (ii) a plasmid containing variant genes of the clamp-loader locus. Upon infection, the clamp-loader locus recombines into the T4$^{del}$ genome, and this genome is replicated by the variant of the clamp and clamp-loader genes present in each cell. Variants with reduction in function in the clamp-loader activity will produce fewer phage particles relative to the variants with wildtype-like activity. (**C**) Workflow for comprehensive assessment of fitness effects of all possible single amino-acid mutants of the sliding clamp and the clamp-loader subunits. Codons corresponding to each amino acid are individually mutagenized to NNS (N is a mixture of all four nucleotide bases, S is a mixture of G and C) by PCR, combined in equimolar ratios and transformed into *E. coli* for cloning.

*2017*; *Tondnevis et al., 2016*; *Turner et al., 1999*). Second, the clamp-bound clamp loader complex recognizes and binds to primer-template junctions (*Simonetta et al., 2009*). Finally, DNA recognition by the clamp-loader results in cooperative ATP hydrolysis and release of the closed clamp on DNA (*Liu et al., 2017*; *Marzahn et al., 2014*).

The base of each ATPase domain of a clamp-loader complex contains a loop that binds to a pocket on the surface of the sliding clamp (*Bowman et al., 2004*; *Gulbis et al., 1996*; *Kelch et al., 2011*). Pseudo-symmetry in the structure of the clamp allows all subunits of the clamp loader to engage a clamp simultaneously, using essentially the same set of interactions. Because of the spiral configuration of the ATP-bound AAA+ modules of the clamp loader, the interaction with the sliding clamp results in the conversion of the normally closed and planar clamp into an open spiral form (*Kelch et al., 2011*; *Miyata et al., 2005*). A key mechanism that enables the clamp loader to play a matchmaker role is the suppression, in the absence of DNA, of the ATPase activity of the clamp loader by the clamp (*Turner et al., 1999*). The recognition of primer-template junctions causes the rearrangement of the ATPase subunits into catalytically competent configurations. This triggers cooperative ATP hydrolysis, which weakens the spiral organization of the ATPase modules and causes the release of the closed clamp on DNA (*Kelch et al., 2012*).

The T4 bacteriophage (T4) has been an important model system for establishing many fundamental principles of DNA replication (*Alberts et al., 1983*; *Benkovic and Spiering, 2017*). T4 encodes its own replication proteins, including a DNA polymerase, a helicase and a clamp loader/clamp system. Like eukaryotic PCNA, the T4 sliding clamp, gene product 45 (gp45), is a trimer (*Moarefi et al., 2000*; *Shamoo and Steitz, 1999*). The subunits of the T4 clamp loader resemble those of eukaryotic clamp-loader complexes (the Replication Factor C, or RFC, complex), and are encoded by two genes, gene 44 and gene 62 (*Jarvis et al., 1989b*; *Jarvis et al., 1989a*). Gene 44 encodes the ATPase subunit (gene product 44, or gp44) which occupies positions B, C, D, and E in the assembled pentameric clamp loader (*Kelch et al., 2011*). Position A is occupied by the protein gp62 (gene product 62), encoded by gene 62. The AAA+ module of gp62 has degenerated and is no longer recognizable as such. Instead, the remnants of the AAA+ module, as well as an additional domain attached to the collar, function as a clasp that connects the AAA+ module of the B subunit to that of the E subunit (*Figure 1A*). We will refer to gp62 as the 'clasp' subunit of the clamp loader.

The AAA+ family, of which the clamp loaders represent an important clade, are involved in the remodeling of many macromolecular complexes, and also function as motors that transport cargo within the cell (*Gates and Martin, 2020*; *Schmidt and Carter, 2016*). Recent structural studies of AAA+ family members have revealed a common theme – like the clamp loaders, the ATPase modules of AAA+ assemblies form a spiral around a central axis that is occupied by the substrate to be remodeled, or the structural element on which mechanical work is performed (*Erzberger and Berger, 2006*; *Gates and Martin, 2020*). This reflects an ancient origin of the AAA+ helicases from oligomeric RNA helicases that were present in the last universal common ancestors of all life (*Iyer et al., 2004*; *Mulkidjanian et al., 2007*). The ATPase domain of a AAA+ module (referred to as Domain 1, see *Figure 1A*) belongs to the much broader class of P-loop NTPases that also includes monomeric GTPases and oligomeric ATPases of the RecA and F-type ATPase families (*Erzberger and Berger, 2006*; *Iyer et al., 2004*). These proteins share evolutionarily ancient Walker A and Walker B sequence motifs (*Cullmann et al., 1995*; *Guenther et al., 1997*; *Longo et al., 2020*; *Neuwald et al., 1999*). Residues in the Walker A motif (residues 45–58 in the ATPase subunit of T4 clamp loader) form the P-loop that cradles the β and γ phosphate groups of the nucleotide. The Walker B motif (residues 103–108 in T4 clamp loader) includes two acidic residues that coordinate the $Mg^{2+}$ ion that is necessary for nucleotide binding (*Chiraniya et al., 2013*; *Seybert and Wigley, 2004*).

AAA+ proteins differ from other oligomeric ATPases in the location of the so-called 'arginine finger', a residue in Domain 1 that is presented by one subunit (e.g. subunit C in the clamp loader) and senses the terminal phosphate group of ATP bound by the previous subunit (e.g. subunit B, see *Figure 1C*; *Wendler et al., 2012*). In AAA+ proteins, the arginine finger is presented by the C-terminal end of an α-helix, helix α6, that is analogous to the Switch 2 element of Ras. A loop at the N-terminal end of helix α6 presents a residue that senses the presence of the terminal phosphate of ATP bound to the same subunit (this residue, referred to as Sensor 1, is analogous to Gln 61 in human Ras proteins, which is mutated frequently in cancer). Thus, helix α6 and the preceding loop directly link the ATP bound to one subunit (the proximal ATP) to the ATP bound by the previous subunit (the distal ATP). This connection is likely to be important for the coordinated hydrolysis of ATP at multiple ATP-binding sites in the clamp loader. Another feature that distinguishes AAA+ ATPases from other oligomeric ATPases is the presence of a helical domain, denoted Domain 2, in the AAA+ module (*Figure 1A*). Domain 2 presents another arginine residue (Sensor 2) that

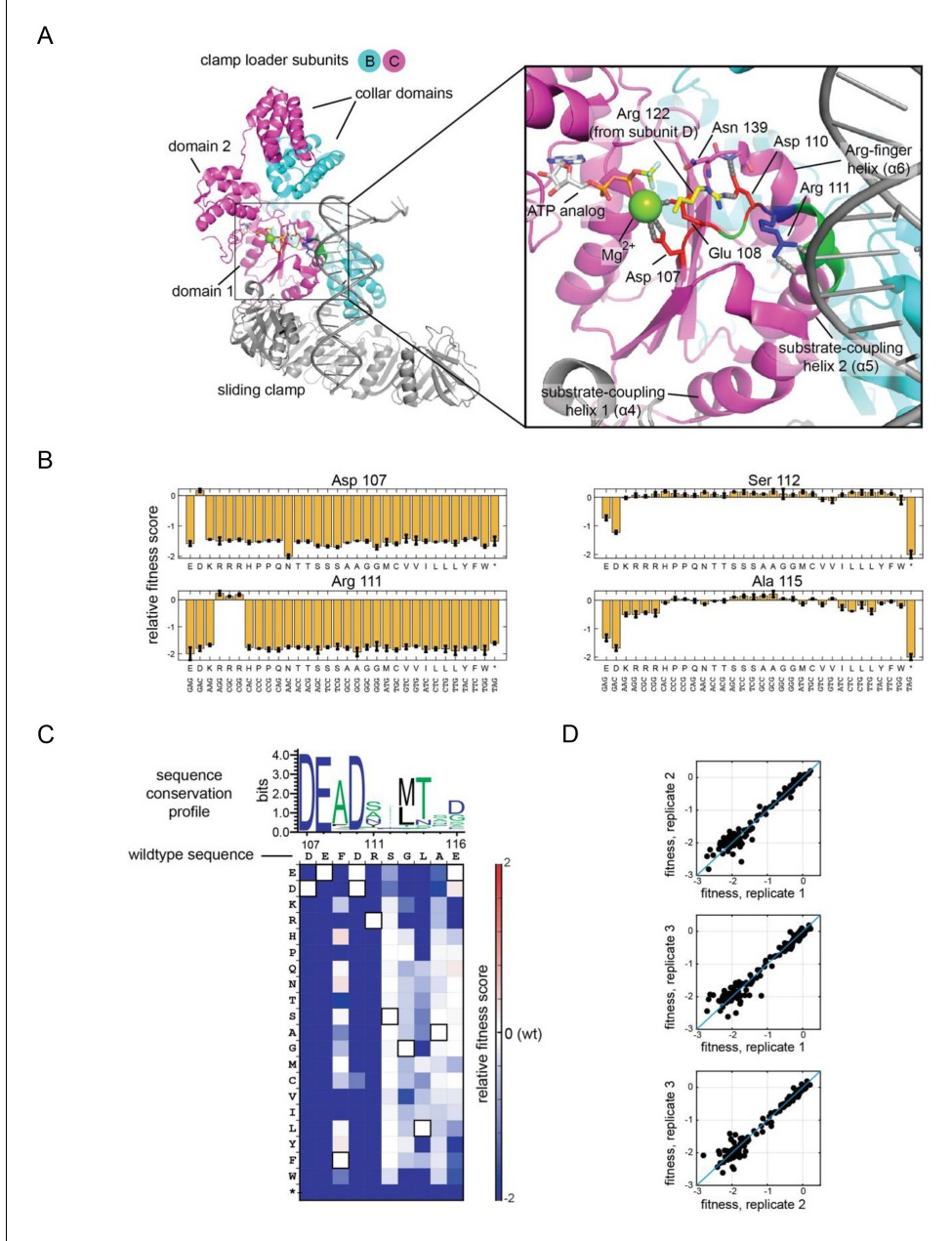

**Figure 3.** Validation of the high-throughput phage-propagation assay with a library targeting the 10-residue region including the Walker B motif of the ATPase subunit. (**A**) The location of the 10-residue region in the structure of the clamp loader. (**B**) Relative fitness values of all 32 codons from the NNS substitutions at mutationally sensitive positions (107 and 111) and mutationally tolerant positions (112 and 115). (**C**) Relative fitness values for each amino acid substitution in the 10-residue region depicted as a heatmap, with wildtype-like fitness as white (score of 0), loss of function as shades of blue and gain of function as shades of red. Pixels corresponding to the amino acid present in the wild-type sequence at each position are outlined in black. Sequence logo generated from 1000 related phage ATPase sequences, with the sequence of T4 bacteriophage shown above the heatmap for reference. (**D**) Agreement between fitness measurements from three replicate experiments, where each point in the scatter plot represents the fitness measurements made from two trials.

The online version of this article includes the following figure supplement(s) for figure 3:

**Figure supplement 1.** Biochemical measurements for selected mutants of the ATPase subunit.

coordinates ATP bound to Domain 1, making the inter-domain conformation of the AAA+ module sensitive to the presence of ATP.

The sequences of sliding clamps and clamp loaders display a high degree of variation across the different branches of life, and even within individual branches such as the T4-like bacteriophages. To what extent does the mutational sensitivity of the system in a particular organism, carrying out the essential function of DNA replication, reflect the sequence diversity seen across the spread of life? The clamp loader subunits respond cooperatively to the clamp, ATP and DNA. How do the mechanisms underlying this cooperativity impose constraints on the sequence? Are there residues that are highly sensitive to mutation, but do not have an obvious role in catalysis or molecular recognition? The depth of structural and biochemical information available for clamp-loader complexes make them particularly suitable for the application of deep-mutagenesis methods aimed at addressing these questions (*Boucher et al., 2016*; *Boucher et al., 2014*; *Fowler and Fields, 2014*; *Shah and Kuriyan, 2019*).

Motivated by these questions, we developed a platform to apply deep mutagenesis and high-throughput functional screening to the replication proteins encoded by T4 bacteriophage. The platform enables the mutational sensitivity of these proteins to be measured in a proper biological context, where the fitness of a particular variant is quantified in terms of the replicative fitness of the mutant bacteriophage. We used this phage-propagation assay to map the effects of saturation mutagenesis of the clamp loader and the sliding clamp. The mutagenesis screen identifies functionally critical residues in the protein, including those in the catalytic center and at the interfaces with DNA and the clamp. Consistent with the diversity observed in clamp loader and clamp sequences across all branches of life, we find that the bacteriophage T4 versions of these proteins possess a high tolerance to mutations. There is evidence for extensive epistasis in the ATPase subunits, however, so that the sequence conservation profiles are not good predictors of the detailed mutational sensitivity of the T4 proteins. Importantly, we identify a mutationally-sensitive glutamine residue (Gln 118) whose significance had not been appreciated previously. This residue forms a hydrogen-bonded junction between two α-helices that are part of a structural unit that we term the central coupler, which connects ATP to DNA, the clamp and the ATP bound at neighboring subunits. Gln 118 completes a hydrogen-bonded network that extends from one ATP-binding site to the other, across all the ATPase subunits of the clamp-loader complex. The glutamine-mediated junction, whose importance we have pinpointed in the clamp loader, is also conserved in diverse AAA+ proteins, suggesting that the hydrogen-bonded network is an ancient mechanism that couples ATP hydrolysis to structural transitions and force generation.

## Results and discussion

### Strategy for deep mutagenesis of T4 phage replication proteins

We developed an assay for T4 replication that enables the high-throughput functional screening of variants of any of the phage replication proteins (*Figure 2*). One component of the platform is a modified T4 bacteriophage – termed T4$^{del}$ – in which genes for the replication proteins of interest are deleted. This modified phage can infect *E. coli* successfully, but cannot replicate inside the host. The second component of the platform consists of *E. coli* cells that carry plasmids encoding variants of the replication genes that are deleted in T4$^{del}$. The expression of the replication proteins in *E. coli* is under the control of natural T4 promoter sequences in the plasmids, and so the protein levels in this experiment are expected to be approximately what they would be during a normal phage infection. When T4$^{del}$ phage infect these *E. coli* cells, they are able to replicate their genome by utilizing the proteins encoded by the bacterial plasmid.

When T4$^{del}$ phage infect a library of these modified *E. coli* cells – with individual cells in the library carrying one particular variant of the replication gene under study – the degree to which the phage in a given cell are able to replicate depends on the functional fitness of the replication protein variant encoded by the plasmid in that cell. Thus, *E. coli* cells expressing wildtype-like, fully functional, variants of the replication genes will produce a large number of T4$^{del}$ phage particles, whereas cells expressing functionally compromised variants will generate fewer T4$^{del}$ phage particles.

The two components described above allow phage propagation, but this system by itself does not allow us to score the functional fitness of variants by sequencing the genome of phage that are

generated by replication. The newly-generated T4$^{del}$ phage particles would all be identical to the T4$^{del}$ phage used to start the infection, because the T4 replication genes that enabled phage propagation are present in the bacteria rather than in the phage genome. In order to measure replication fitness by sequencing the genomes of the propagated phage particles, we further modified the host cells by adding a CRISPR-based genome engineering module. This module efficiently recombines the replication genes from the bacterial plasmid back into the genome of T4$^{del}$ phage particles generated in the assay. Sequencing the propagated T4$^{del}$ phage then allows us to score the replication fitness of the variants. This strategy enables us to test variants of replication genes for function in a high-throughput manner, and in its proper biological context. The platform also enables the implementation of in-vitro evolution studies of the replication proteins, although that is not an application discussed in this paper.

## Implementation of the high-throughput assay to study the clamp and the clamp loader

We created a version of the T4$^{del}$ phage in which the clamp gene (gene 45) and the clamp-loader genes (genes 44 and 62, encoding the ATPase and the clasp subunits, respectively) are deleted (*Figure 2A*). We generated a plasmid bearing a copy of the T4 clamp and clamp-loader genes such that T4$^{del}$ can be propagated in *E. coli* cells carrying this plasmid. We made two modifications to the bacterial system to aid with recombination of the clamp and clamp-loader genes into T4$^{del}$. First, we introduced a plasmid carrying genes for CRISPR-cas12a (*Safari et al., 2019*) and a guide RNA that can program cas12a to target an engineered cas12a recognition site in T4$^{del}$ phage. On infection by T4$^{del}$, the host cell can now utilize the cas12 enzyme to cleave the invading genome. The second modification was to the plasmid bearing the clamp and clamp-loader genes: we padded the genes with ~1 kb arms of homology to the clamp and clamp-loader gene locus, to enable recombination of the plasmid-borne genes into T4$^{del}$.

To assay the fitness of variants, we first generate a library of *E. coli* cells carrying plasmid-borne copies of the variant T4 replication genes (*Figure 2B*). This library of cells is infected with T4$^{del}$ phage at a multiplicity of infection of 0.001 (MOI, ratio of phage particles to bacterial cells), and phage are collected once all the bacteria are lysed. Illumina sequencing is used to measure the frequencies of the variant DNA sequences in the plasmid – the 'input' population, and the frequency of variant DNA sequences in the recombined T4$^{del}$ phage, the 'output' population. We define the fitness, F$^{(i)}$, of a variant allele, *i*, to be the fitness of the variant relative to the wild-type gene, as follows:

$$\mathrm{F}^{(i)} = log_{10}\left(\frac{\mathrm{f}_{\mathrm{output}}^{(i)}}{\mathrm{f}_{\mathrm{input}}^{(i)}}\right) - log_{10}\left(\frac{\mathrm{f}_{\mathrm{output}}^{\mathrm{WT}}}{\mathrm{f}_{\mathrm{input}}^{\mathrm{WT}}}\right)$$

where $\mathrm{F}^{(i)}$ is the fitness score for variant $i$, $\mathrm{f}_{\mathrm{output}}^{(i)}$ is the number of counts observed for variant $i$ in the recombinant-phage library, $\mathrm{f}_{\mathrm{input}}^{(i)}$ is the number of counts observed for variant $i$ in the parent library, with the 'WT' superscripts representing counts for the wildtype sequence. A variant with a fitness score of zero propagates at the same rate as the wild-type phage. Variants with fitness scores of +1 and −1 propagate 10-fold faster and slower, respectively, than wild-type phage.

## Validation of the high-throughput assay

To assess the robustness of the phage-propagation assay we carried out saturation mutagenesis of a 10-residue segment of the ATPase subunit of the T4 clamp loader (residues 107–116 of gp44; *Figure 3A*). This segment contains several residues that are critical for function, including a part of the Walker B motif, and these are expected to be extremely sensitive to mutation. We used 'NNS' codons (N: A, C, T or G; S: C or G) to introduce 32 possible codons, coding for each of the 20 amino acids with redundancy, at each position within this 10-residue segment. This led to the generation of a library of 320 variants of the ATPase subunit, which was used to generate recombinant T4$^{del}$ phage that were scored for fitness based on phage propagation efficiency.

The fitness scores for point mutants in this segment are represented in two ways. First, the fitness scores for the 32 individual codon substitutions at four representative positions are shown as bar graphs (*Figure 3B*). The scores are on a logarithmic scale, and so scores of −1 and −2 correspond

to 10-fold and 100-fold reductions in sequence counts for a particular variant relative to the wild-type sequence. Second, the mean fitness scores for each amino acid substitution, averaged over redundant codons, are shown in a heatmap (*Figure 3C*). The wild-type protein sequence is represented along the horizontal axis and the 20 amino acids are represented along the vertical axis.

Two aspects of these results establish that there is internal consistency in the data. First, there is a high degree of reproducibility across three independent trials, indicating that the stochasticity of plasmid recombination events does not affect the fitness measurements significantly (*Figure 3D*). Second, the data reveal excellent agreement between the fitness effects of variants encoded by synonymous codons (*Figure 3B*). For example, while position 111 can only tolerate an arginine residue, all three arginine codons at this position result in fitness scores near zero, corresponding to phage propagation rates that are near wildtype, and all other substitutions are extremely detrimental (fitness scores less than −1.7, a 50-fold reduction in abundance relative to wild-type phage).

The phage-propagation assay is initiated with only a small fraction of host cells that are infected (~0.1% of cells are infected at MOI of 0.001). The emerging recombinant phage will further infect previously uninfected host cells, until all host cells are lysed. Since an infected cell produces ~250 new phage particles (*Rabinovitch et al., 1999*), recombinant phage can cause up to two subsequent waves of infection. For the conditions used in our experiments, the reproducibility in fitness measurements for the library of single amino-acid variants in the ten-residue segment, at the codon level, indicates that the noise introduced by multiple cycles of phage infection is not a serious issue.

Residues within the ten-residue segment that are expected to be functionally critical are highly sensitive to mutation, providing additional reassurance that the assay is reporting faithfully on replication fitness. The first four residues of this segment (residues 107 to 110, with sequence DEFD) is part of the Walker B motif, and corresponds to the DExD/H motif that is found in many helicases (this motif is DExD in eukaryotic, viral, and archaeal clamp loaders). Asp 107 and Glu 108, at the first two positions of the DExD motif, coordinate the $Mg^{2+}$ ion bound to ATP. Glu 108 has an additional catalytic role in activating the water molecule that attacks the terminal phosphate group of ATP (*Ogura and Wilkinson, 2001*). Consistent with their crucial functions, these two residues do not tolerate any substitution in the phage-propagation assay. Even the conservative substitutions of Asp 107 by glutamic acid, or Glu 108 by aspartic acid, result in fitness scores corresponding to at least a 200-fold reduction in the number of output phage.

Asp 110, the fourth residue in the DExD motif, is likely to play a regulatory rather than a catalytic role. Asp 110 forms an ion pair with the sidechain of Arg 122 in the adjacent subunit when the clamp loader is bound to DNA (*Kelch et al., 2011*), but this interaction is disrupted in a way that blocks ATP hydrolysis when DNA is not bound, an important point that we return to later (*Bowman et al., 2004*; *Gaubitz et al., 2020*). The mutational data in *Figure 3* show that Asp 110 is highly sensitive to mutation, with only Asp and Glu being tolerated. Arg 111 is a DNA-interacting residue that is extremely sensitive to any substitution in the phage replication assay. Even the conservative replacement of this arginine by lysine results in a greater than 200-fold decrease in phage propagation.

We purified six mutant clamp loaders and carried out biochemical assays to assess the effect of the mutations on function. The six variants that were analyzed had fitness scores spanning the range of observed values. Biochemical assays cannot account for the complexity of phage propagation, and so they are potentially limited in their ability to validate the results of the phage-propagation assay. Nevertheless, our experiments suggest that a reduction of fitness in the phage-propagation assays corresponds to a loss of efficiency in the clamp-loading reaction and in phage replication.

In one biochemical assay, we measured the rate of DNA-stimulated ATP hydrolysis catalyzed by the clamp loader in the presence of clamp and primed DNA (see Methods) (*Goedken et al., 2005*; *Figure 3—figure supplement 1A*). The wildtype clamp loader and the Q171A mutant (F = 0.0) both have high ATPase activity. Four variants of the ATPase subunit (R151E, F109V, N139I, and R111S) that are mutationally sensitive in the phage-propagation assay have low ATPase activity in the in vitro assay.

In a second assay, we measured the ability of the mutant clamp loaders to support replication of single-stranded phage M13 DNA, in the presence of the sliding clamp, DNA polymerase and accessory proteins (*Figure 3—figure supplement 1B*; *Seville et al., 1996*). In the time allowed for the reaction to proceed (10 min), clamp loaders with high fitness scores in the phage-propagation assay (the wildtype clamp loader and the Q171A variant) are able to convert all the single-stranded DNA to double-stranded DNA. As expected, clamp-loader variants with low fitness scores (the R151E,

R111S and N139I variants) were unable to take DNA replication to completion. Despite a low fitness value for the F109V variant, it performs like the wildtype in the replication assay. Phage bearing the F109V mutation cause much smaller plaques to form in *E. coli* colonies, compared to the wild-type variant, confirming that the mutation does lead to a reduction in fitness. We suspect that the wild-type-like performance of this variant in the in vitro replication assay is due to the long reaction times used in the assay (10 min).

## Deep mutagenesis of the clamp-loader system

We carried out saturation mutagenesis of the sliding clamp (gp45, 227 residues), the ATPase subunit of the clamp loader (gp44, 318 residues) and the clasp subunit of the clamp loader (gp62, 186 residues) (*Figure 4*). Every residue in these proteins was substituted by all other amino acids, one at a time, and the fitness scores of the resulting variants were determined using the phage-propagation assay. We partitioned each gene into overlapping segments of ~450 nucleotides each for the mutational scans, so that the mutated regions are covered reliably by sequencing with a read length of 500 nucleotides on the Illumina MiSeq sequencing platform (see Methods for details). This resulted in two segments each for the sliding clamp and the clasp subunit of the clamp loader, and three segments for the ATPase subunit. We generated libraries of variants for each of the segments, for a total of seven plasmid pools, that we then subjected separately to selection with the phage-propagation assay.

We performed the selection in triplicate for each pool. For each trial of the assay, bacterial cells were transfected with plasmids from one of the pools, and ~$10^{10}$ of these cells were infected with ~$10^7$ T4$^{del}$ phage particles (MOI of 0.001) at 37°C and incubated overnight (see Methods for details). We purified plasmid DNA from the bacteria just before phage infection to use as the 'input' library for sequencing, and used the phage particles that were generated overnight from the infected culture to produce the 'output' library. We amplified the mutated regions of the genes of interest by PCR and sequenced the regions using a MiSeq sequencer.

We note that the data obtained from the larger-scale mutagenesis of proteins in the clamp-loader complex are noisier than the data obtained for the small library created for saturation mutagenesis of the 10-residue Walker-B segment, discussed above. The reason for the increased noise is the uneven distribution of variants in the starting library, a consequence of using degenerate mutagenic primers that also include primers with a perfect match to the PCR template. As an example of the skewed distribution of the variants, consider the counts observed for each of the variants in the input library of the second pool of the ATPase subunit, covering residues 117–230. Of the 1,356,540 total reads for this library, the wild-type gene accounted for 517,908 counts or ~38%, and 326 variants (13.6% of the variants in the pool) had fewer than 50 counts. The data obtained from these libraries provide reliable information when considering residue-averaged mutational effects, as judged by the reproducibility of the data. If, however, one is interested in the fitness cost of a specific substitution, then the use of alternate methods of library construction, or more focused libraries with a balanced distribution of variants, as was done for the 10-residue segment discussed earlier, will be important to obtain the most accurate data.

The distribution of fitness scores for the point mutants of the three mutated proteins are shown in *Figure 4*. We divide the mutations into two categories – those corresponding to severe loss of function, with fitness scores of less than −1, and those with moderate to neutral effects, with fitness scores in the range of −1 to +1. The two non-enzymatic proteins in the complex – the sliding clamp and the clasp subunit of the clamp loader – have mutational effects that are largely confined to the moderate to neutral category. Within the ATPase subunit of the clamp loader, the collar domain also has mutations that lie within the moderate to neutral category. Residues that are highly sensitive to mutation are limited mainly to the AAA+ module of the ATPase subunit of the clamp loader.

It is striking that the sliding clamp is so tolerant of mutation compared to the clamp loader (*Figure 4—figure supplement 1*). This result is consistent with the fact that although the architecture of the clamp is conserved in all the branches of life, no position in the sliding clamp is strictly conserved in terms of sequence, even when considering sliding clamps from T4-like bacteriophages alone. The residues in the sliding clamp that have the largest mutational effects are buried hydrophobic residues. In particular, the solvent accessible surface area of a residue in the structure of the clamp is a good predictor of the mutational tolerance of the residues, with surface-exposed residues more tolerant of substitution than buried residues. This suggests that mutational sensitivity in the sliding

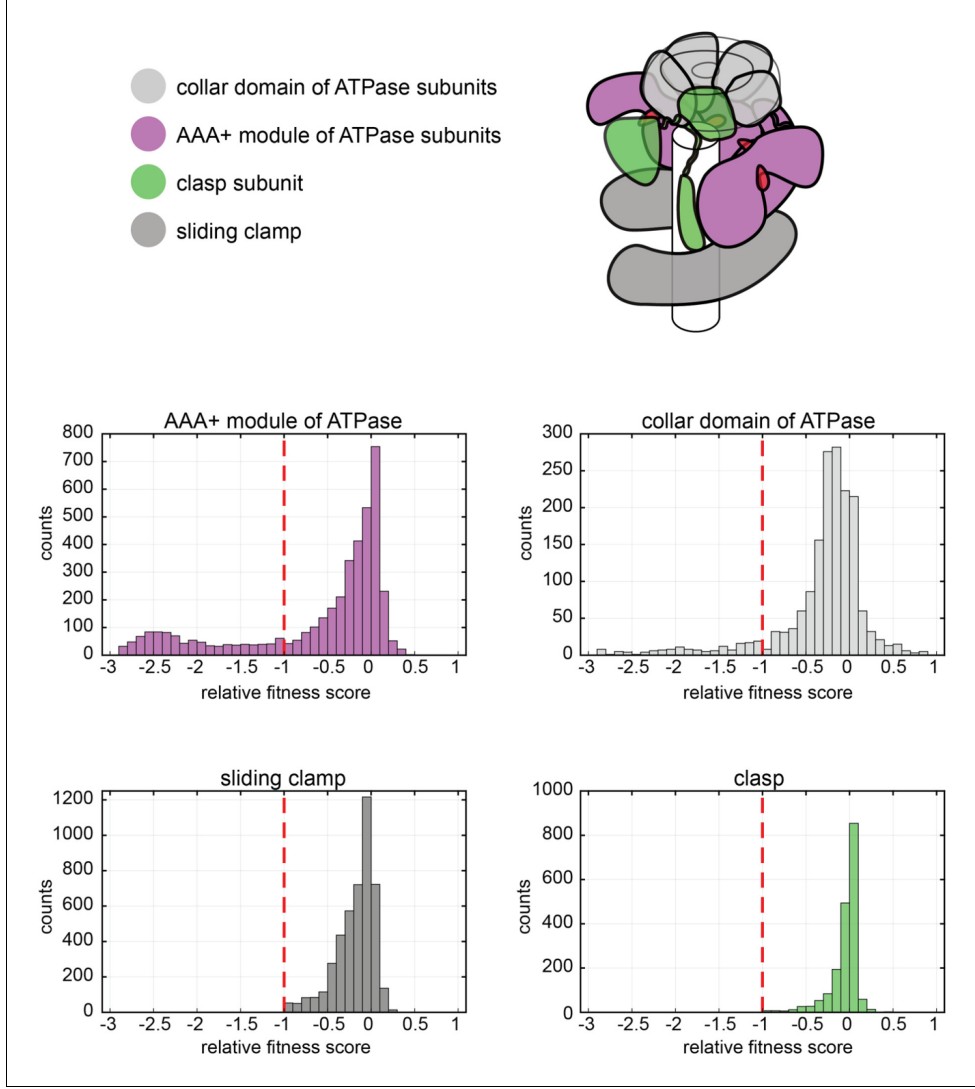

**Figure 4.** Distribution of relative fitness values from deep mutagenesis of the clamp loader system. The histograms show the spread of relative fitness values for AAA+ module and collar domain of the ATPase subunit, the clasp subunit and the sliding clamp.

The online version of this article includes the following figure supplement(s) for figure 4:

**Figure supplement 1.** Mutational sensitivity of the T4 sliding clamp (gp45).

**Figure supplement 2.** Mutational sensitivity of the clasp subunit (gp62) of the T4 clamp loader.

**Figure supplement 3.** Mutational sensitivity of the collar domain (residue 231–319 of gp44) of the ATPase subunit.

clamp is determined largely by the need to preserve the integrity of the protein fold, a result that is in keeping with topological, rather than specific, interaction of the sliding clamp with DNA. These results also suggest that the functionally important interactions made by the sliding clamp with DNA (*De March et al., 2017*; *McNally et al., 2010*), the clamp loader (*Bowman et al., 2004*), and the DNA polymerase (*Shamoo and Steitz, 1999*) are not affected significantly by single mutations in the sliding clamp.

The mutational scan of the clasp subunit of the clamp loader (gp62, the A subunit of the complex) also reveals a high degree of mutational tolerance, with sensitivity to mutation generally being consistent with the demands of maintaining the protein fold (*Figure 4—figure supplement 2*). However, the two residues in the clasp that are the most sensitive to mutation, Trp 17 and Trp 111, have obvious importance in clamp loader function, rather than in folding. Trp 17 packs against two critical α helices, denoted α4 and α5, in the adjacent B subunit of the clamp-loader complex – the

importance of these helices is discussed below (*Figure 1C*). The sidechain of Trp 111 buttresses the first two unpaired bases of the template strand that are peeled away from the primer strand, and this interaction is presumably important for the ability of clamp loaders to load clamps specifically at primer-template junctions. Thus, despite the general tolerance to mutation of the clasp subunit, the clear functional relevance of the two most sensitive sites in the subunit provides further reassurance that the phage-propagation assay is a robust reporter of the efficiency of clamp-loader function in DNA replication.

## Analysis of mutational sensitivity in the AAA+ module of the T4 clamp loader

The mutational data for the clamp loader provide an opportunity to analyze the extent to which sequence conservation across an evolutionarily-related protein family correlates with the effects of mutations on fitness in a specific member of the family. The sliding clamp, the clasp subunit of the clamp loader and the collar domain of the ATPase subunits are very divergent in sequence, even when considering only phage sequences, making the results of such a comparison difficult to interpret. We therefore restricted our analysis of the correlation between sequence variation and mutational fitness to the AAA+ module.

*Figure 5* shows a heatmap corresponding to results of the deep mutagenesis analysis for the AAA+ module of the T4 clamp loader. For comparison, the sequence-conservation profile for the AAA+ modules of a set of bacteriophage clamp loaders is shown above the heat map. We generated this conservation profile by aligning 1000 sequences of clamp-loader ATPase subunits from bacteriophage genomes, and the results are represented as a sequence logo (*Crooks et al., 2004*) in *Figure 5*. The height of the single-letter codes for an amino acid at a particular position in the sequence logo is proportional to the sequence conservation at that position.

Visual comparison of the mutational profile and the sequence-conservation logo shows that regions of high sequence conservation in the phylogenetic analysis correspond to regions of the T4 AAA+ module that are mutationally sensitive in the phage-propagation assay. Likewise, regions that are variable in the sequence-conservation profile are mutationally tolerant in the phage-propagation assay. For a more quantitative analysis, we reduced the profile of mean fitness effects at each position to a binary descriptor in which residues are denoted as either tolerant of mutation, or not. Analysis of the false-positive rate for prediction of the fitness effects based on the sequence conservation scores of each residue shows that the positional conservation scores are good predictors for whether or not a residue is sensitive to mutagenesis (*Figure 4—figure supplement 1A*).

It is important to recognize, however, that the phylogenetic conservation profile is not a good predictor of the actual effect on fitness of specific mutations in the T4 clamp loader. For example, considering the 10-residue segment that was analyzed in detail earlier, the sequence profile shows that the first, second, and fourth residues in the DExD motif are very highly conserved as Asp, Glu, and Asp in clamp-loader sequences (*Figure 3A*). While the first two residues cannot be replaced by any other residue in the mutational data derived from the phage-propagation assay, Asp 110, the fourth residue in the motif, can be replaced by glutamate without loss of fitness (*Figure 3B,C*). As noted earlier, Asp 110 is likely to play a regulatory rather than a catalytic role, and different clamp loaders might display different sensitivity at this site. Another example is provided by Arg 111, the sidechain of which interacts with DNA. The mutational data show that this residue does not tolerate any substitution, even by lysine. In contrast, the sequence logo shows no conservation at this site, and positively-charged residues are not prominent in the sequence logo.

These results point to an important distinction between the sequence-conservation data and the mutational-fitness data. The latter reflect the idiosyncratic epistatic constraints imposed by the evolutionary history and the specific molecular properties of T4 proteins, and the particular context in which they operate. Although the T4 clamp loader clearly relies on Arg 111 to recognize DNA, other phage clamp loaders have presumably evolved mechanisms for DNA recognition that rely on different residues. A similar effect is seen for the third position (109) of the DExD motif, which is variable in the sequence alignments, with alanine being the residue that is most frequently found at this position in phage clamp loaders. The T4 clamp loader has phenylalanine at this position, and substitution with alanine leads to a reduction in fitness.

We analyzed the extent to which the AAA+ module can tolerate substitutions that represent the most commonly occurring residues in the sequences of other phage clamp loaders. For residues

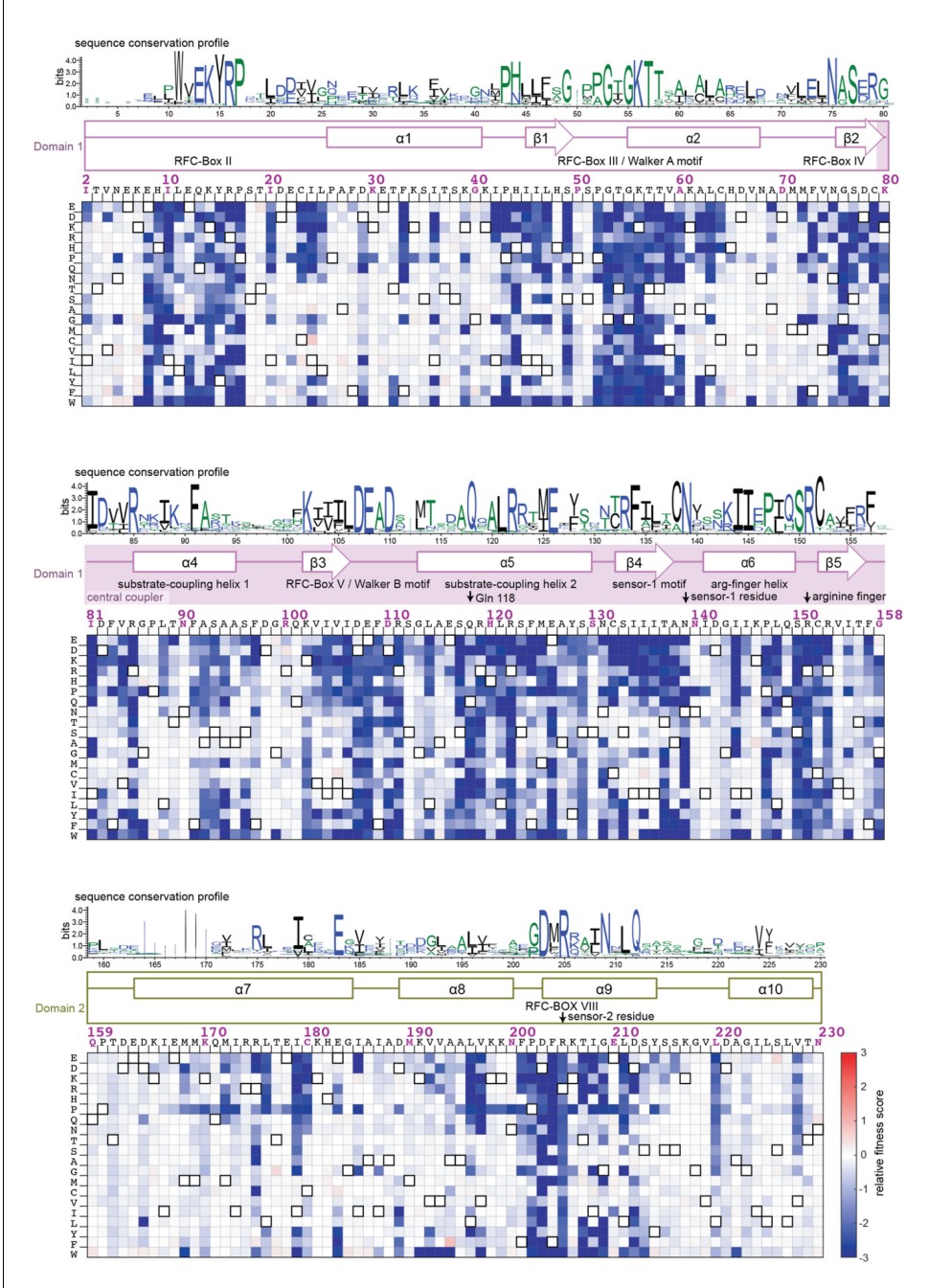

**Figure 5.** Mutational sensitivity of the AAA+ module of the ATPase subunit. The fitness scores for all point mutations to the AAA+ module are shown as a heatmap, as in *Figure 3C*. The secondary structure of the wildtype sequence is indicated above the heatmap, with annotations identifying key elements of the AAA+ module (*Cullmann et al., 1995*; *Guenther et al., 1997*). The sequence-conservation profile derived from 1000 phage clamp-loader sequences is depicted above the secondary-structural elements as a sequence logo (*Crooks et al., 2004*).

The online version of this article includes the following figure supplement(s) for figure 5:

**Figure supplement 1.** Comparison of mutational sensitivity in the AAA+ module of the ATPase subunit to evolutionary sequence conservation.

where the consensus residue in the sequence alignment is present in the T4 AAA+ module, we considered the fitness score in the T4 mutational data for the second-most frequent residue at that position in the sequence-conservation profile. For positions where the T4 sequence does not have the consensus residue, we considered the fitness score for substituting in the consensus residue. In this manner, we calculated the fitness effect of substituting residues of the AAA+ module with the most likely substitution based on the evolutionary record. We found that the mean fitness effect of these substitutions corresponds to a three-fold reduction in phage production (relative fitness score < −0.5) in the T4 system (*Figure 5—figure supplement 1B*). This analysis is restricted to positions in the sequence alignment that are not the most highly conserved, since for very highly conserved positions the second-most frequent residue is not well determined.

## The residues that are the most mutationally sensitive form a contiguous belt that links DNA, ATP, and the sliding clamp

We examined the spatial proximity of the residues that are the most sensitive in the mutational screens. These residues admit few substitutions, or none, without a substantial reduction in fitness, and have mean mutational fitness scores below −1.6. As shown in *Figure 6A*, there are 18 such residues in each subunit, and they form a contiguous belt in the interior of the clamp loader complex, spanning the four ATPase domains and connecting each ATP-binding site to DNA and to the ATP molecules bound at adjacent subunits. The majority of these residues are clustered around the interfacial sites between neighboring subunits, where ATP is bound (*Figure 6B*). At each such site, 11 of the mutationally-sensitive residues are provided by the subunit to which the ATP is bound, and three, including the arginine finger (Arg 151), are provided by the adjacent subunit (*Figure 6B*). From this central cluster, the mutationally-sensitive residues form a connected network that extends out to two arginine residues that interact with DNA, and to two other residues that are at points of interfacial contact between the subunits.

All but three of the residues in the most sensitive set have polar or charged sidechains that form hydrogen bonds. The three exceptions are all glycine residues that have specific structural roles, including two in the P-loop (Walker A motif). Hydrogen-bonding interactions are exquisitely sensitive to distance and geometry, explaining the extreme sensitivity to mutation of these residues. Interestingly, the set of most-sensitive residues does not include any residues that make contact with the sliding clamp. The interaction between the clamp loader and the clamp involves hydrophobic contacts, which are not very sensitive to geometrical details. For example, a key contact between the clamp loader and the clamp involves the sidechain of Phe 97 in the ATPase subunit of the clamp loader. Phe 97 can be replaced by Cys, Val, Ile, Leu, and Tyr without a large reduction in fitness (*Figure 5*).

## A mutationally-sensitive glutamine sidechain forms a bridge between the sites of proximal and distal ATP coordination

In the set of mutationally-sensitive residues in the AAA+ module, there is only one residue, Gln 118, that does not have an obvious function in ATP or DNA binding, or in interfacial interactions. We confirmed that Gln 118 is highly sensitive to mutation by performing the phage-propagation assay with a focused library of point mutants limited to position 118 (*Figure 6—figure supplement 1*). This analysis shows that the only substitution with a minimal effect on fitness is the replacement of the glutamine by histidine, the sidechain of which can also form multiple hydrogen bonds. The significance of Gln 118 becomes apparent when examining the spatial distribution of the most sensitive residues. Within each subunit, the sensitive residues form two spatially separated clusters, and Gln 118 is the only bridge that connects the two clusters. The principal cluster is organized around the ATP bound to the same subunit (referred to as the *proximal* ATP). The second cluster is located on the surface of the AAA+ module that is distal to the proximal ATP, and this cluster includes the two arginine residues that interact with the ATP bound to the preceding subunit (the *distal* ATP; for example, considering subunit C, the proximal ATP is the one bound to subunit C, and the distal ATP is bound to subunit B, see *Figure 6A*).

The molecular surfaces of the mutationally-sensitive residues are shown in *Figure 6B*, with and without the inclusion of Gln 118. With Gln 118, there is one contiguous cluster of sensitive residues spanning all subunits. Without Gln 118, the mutationally sensitive residues form unconnected

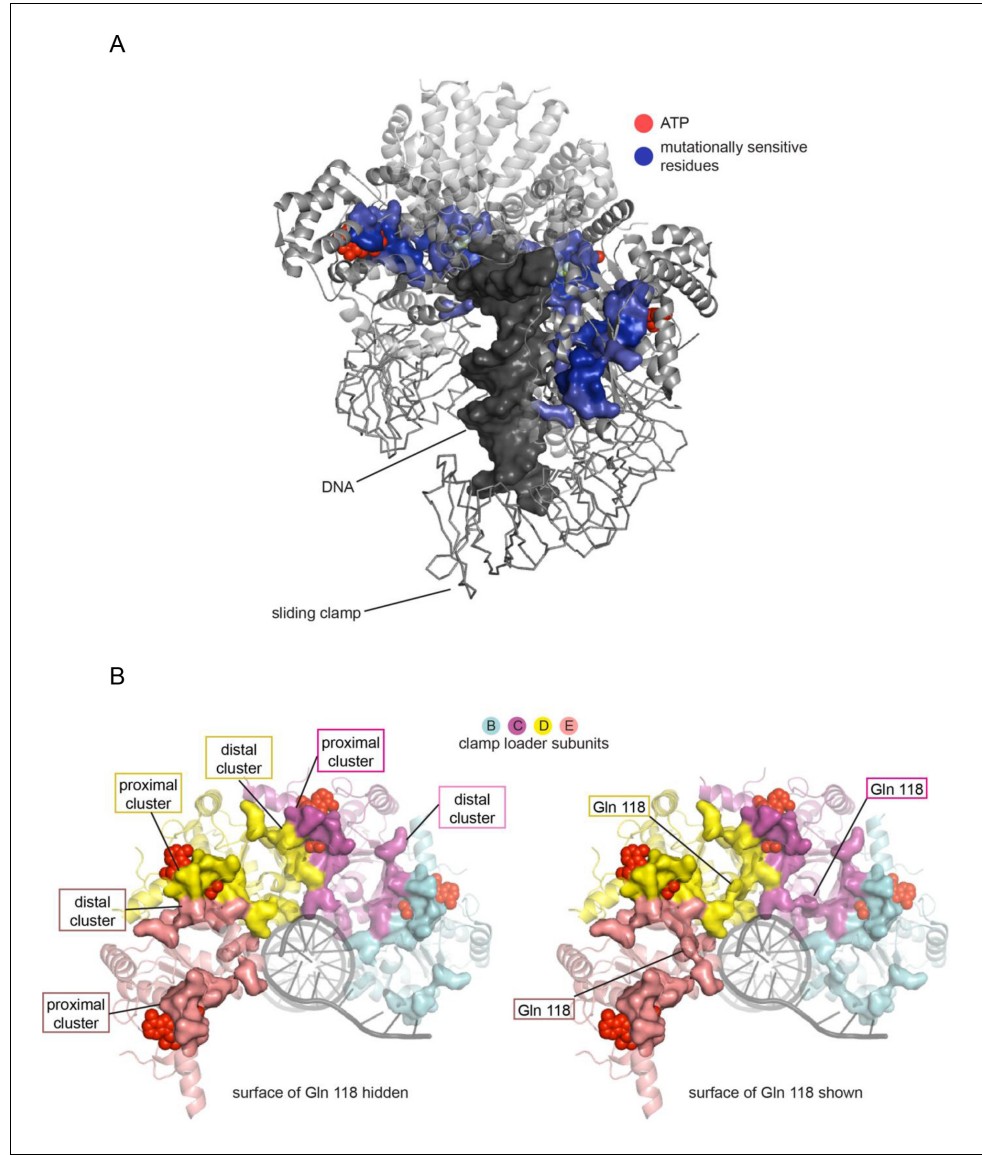

**Figure 6.** Mutationally-sensitive residues form a contiguous network in the structure of the clamp-loader complex, connecting sites interacting with ATP, DNA and the sliding clamp. (**A**) Positions with mean mutational effect <= −1.6 are rendered in surface representation and colored blue. (**B**) Mutationally sensitive residues, viewed from the top (collar domain not shown) without (left) and with (right) Gln 118 shown.

The online version of this article includes the following figure supplement(s) for figure 6:

**Figure supplement 1.** Mutational sensitivity of residue 118 in the ATPase subunit of the T4 clamp loader, tested with a small library of 20 amino acid substitutions.

**Figure supplement 2.** Expression levels of mCherry tagged wildtype and Q118N variant of the clamp loader, observed by flow cytometry.

clusters, centered around each ATP, with two subunits contributing residues to each ATP-binding site. Broadening the definition of mutationally-sensitive residues to include residues that tolerate more substitutions maintains Gln 118 as the unique residue that bridges the two clusters in each sub-unit. The threshold has to be lowered until residues in the hydrophobic core are included before additional residues that form bridges between the distal and proximal clusters are found.

Gln 118 is presented by helix α5. This helix, along with helices α4 and α6, is part of a structural unit that forms the distal surface of Domain 1 of each AAA+ module, which packs against the ATP-binding face of the preceding AAA+ module (see *Figure 1C*). These helices, along with the loops

leading into them, form a contiguous surface in the center of the AAA+ assembly, wrapping around the DNA, making contact with ATP, and docking on the sliding clamp (*Figure 7*). We refer to this unit as the '*central coupler*' (*Figure 1C*), a term suggested by a recent analysis of DnaC, a DNA helicase loader, in which an arginine residue within this unit is referred to as a 'coupler' (*Puri et al., 2020*).

In the central coupler, helix α4 interacts with DNA at one end, and with the sliding clamp at the other end, and is referred to as the substrate-coupling helix 1 (*Figure 1C*). Helix α5 interacts with DNA at its N-terminal end, and is referred to as the substrate-coupling helix 2 (*Figure 1C*). The substrate-coupling helix 2 also presents the sidechain of Arg 122 to interact with the second aspartate of the DExD box of the preceding subunit, in the Walker-B motif. As noted earlier, this interaction has been postulated as being important for coupling DNA binding to ATPase activity (*Gaubitz et al., 2020*). The N-terminal end of the substrate-coupling helix 2 is connected to the strand bearing the DExD box of the same subunit, and so this helix links DExD boxes from one subunit to the next. Helix α6 is referred to as the Arg-finger helix, because it presents the arginine finger to the terminal phosphate group of the ATP bound at the distal site (that is, in the preceding AAA+ module). The N-terminal end of this helix is connected to a loop that connects to Sensor 1, which senses the terminal phosphate group of ATP bound at the proximal site, in the same AAA+ module, putting the Arg-finger helix in a key position for coordinating ATP hydrolysis across subunits.

The central coupler is made up of L-shaped structures because each helix within it has a shorter N-terminal helix or turn that leads into the main helix (*Figure 1C*). The L-shaped architecture of the central coupler is conserved across AAA+ ATPases, and it allows these helices, and the loops preceding them, to wrap around the distal face of Domain 1, allowing interactions with the proximal ATP, with DNA, and with the distal ATP. Gln 118 plays a crucial role in maintaining the structural integrity of the central coupler, because it provides a hydrogen-bonded bridge between the substrate-coupling helix 2 (α5) and the elbow bend that leads into the main segment of the Arg-finger helix (α6) (*Figure 1C* and *Figure 7A*). The interaction is a classical one for glutamine, involving the formation of two hydrogen bonds between the sidechain of glutamine and a carbonyl group and an amide group in the backbone segment of the elbow leading into the Arg-finger helix. In addition, the sidechain of Gln 118 makes a hydrogen bond with the backbone carbonyl group of Phe 109, the third residue in the DExD box of the Walker-B motif. This interaction is likely to be important for stabilizing the conformation of Asp 110 in the DExD box, a residue that makes an important interfacial interaction with Arg 122, as noted earlier (*Gaubitz et al., 2020*).

Gln 118 is conserved in phage and eukaryotic clamp loaders, but there is no corresponding residue in bacterial clamp loaders. A recent analysis of the sequences of bacterial clamp loaders has identified residues that are uniquely present in those clamp loaders (*Tondnevis et al., 2020*), and it will be interesting to see if that analysis provides clues as to how the bacterial proteins compensate for the absence of the glutamine residue.

## Molecular dynamics simulations indicate that the glutamine-mediated junction is important for rigidity in the central-coupler unit and the stability of the catalytic sites

We carried out molecular dynamics simulations that were initiated from the structure of the ATP-loaded clamp loader bound to primer-template DNA and an open clamp (PDB ID 3u60 [*Kelch et al., 2011*]). We ran two sets of simulations, one for the wildtype clamp loader and one for a mutant form in which Gln 118 in each of the four AAA+ modules was replaced by alanine. Each set of simulations included four independent trajectories spanning 400 ns each, for a total of 1.6 μs. Both the wild-type and the mutant complexes were stable in the simulations, with no large-scale changes in conformation. In particular, over the timescale of these simulations the Q118A mutation does not cause global changes in structure.

A difference between the wild-type and mutant simulations becomes evident when we look at the dynamics of the helices in the central coupler. To do this, we calculated inter-residue correlations for each of the trajectories (see Materials and methods for details). To visualize the correlations, we displayed the strength of the correlation between pairs of residues that are in contact by cylinders connecting the Cα atoms of the residues, with the thickness of the cylinder proportional to the strength of the correlation (*Figure 8B,C*). In the simulations of the wild-type clamp loader, there is

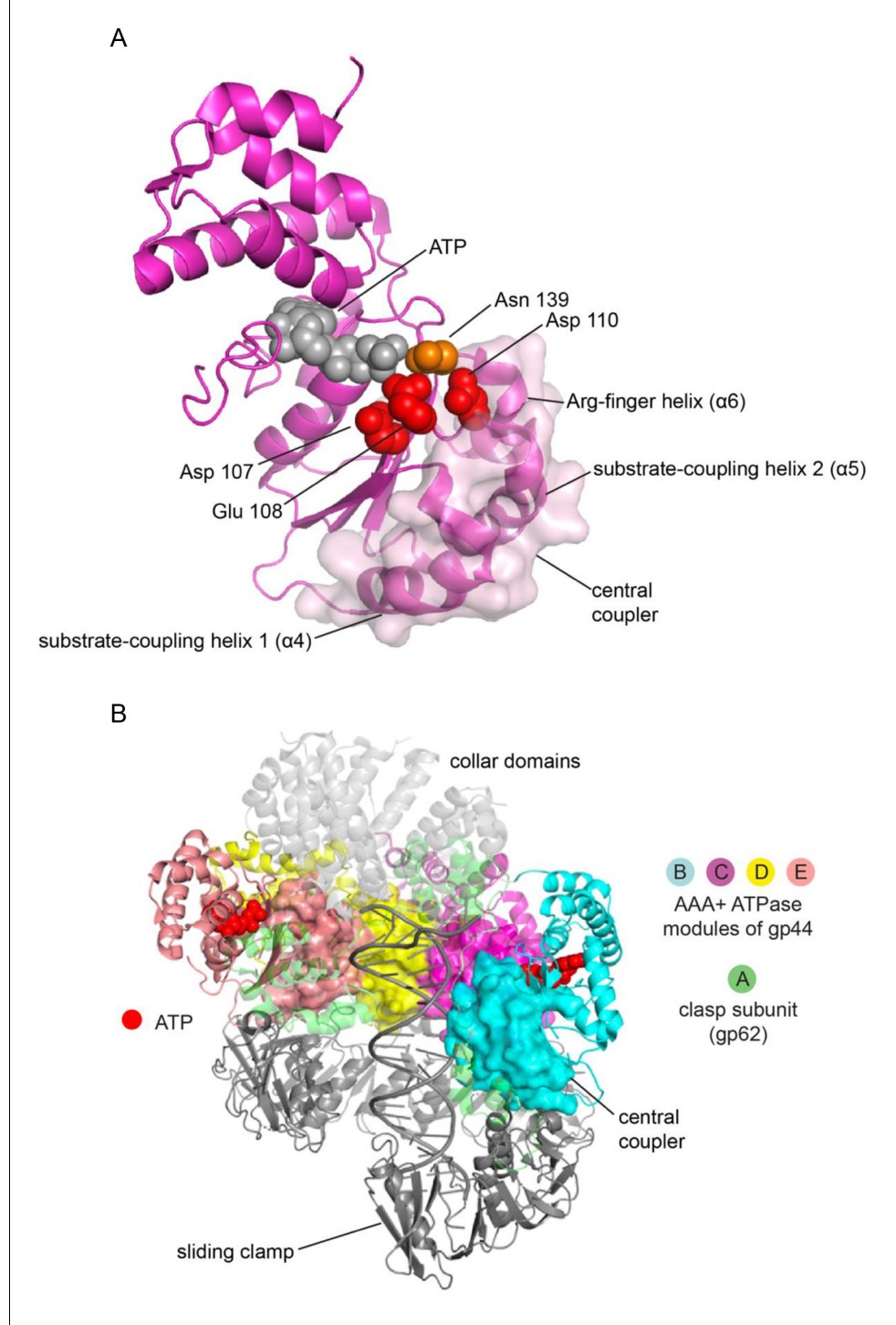

**Figure 7.** The central coupler in the bacteriophage T4 clamp loader. (**A**) The subunit at C position of the T4 clamp loader is shown, with the surface of the central coupler displayed. (**B**) The structure of the complete clamp-loader complex is shown, with the surfaces of the four central-coupler units depicted. (PDB ID: 3u60 *Kelch et al., 2011*).

tight correlation between the substrate-coupling helix 2 of the central coupler and the Arg-finger helix, with Gln 118 serving as the most important node for this correlation. In contrast, in simulations of the Q118A mutant, correlations between the two helices are reduced considerably.

We examined the disposition of the residues in each of the interfacial catalytic centers in the molecular dynamics simulations (*Figure 8D,E*). There are three ATP molecules in each complex,

bound to the B, C, and D subunits – the E subunit has ADP bound to it. For the B and D subunits, we do not observe significant differences in structure and dynamics between the wildtype and mutant simulations. There is, however, a clear difference between the two sets of simulations when we examine the active site that is centered on the ATP bound to subunit C.

A mutational study of the yeast clamp-loader complex (RFC) showed that interfacial coordination of the ATP bound to the C subunit is critical for DNA binding (*Johnson et al., 2006*). Mutation of the arginine finger presented by subunit D to the ATP bound to subunit C results in complete loss of DNA binding, whereas mutation of the other arginine fingers results in much less severe effects (*Johnson et al., 2006*). The particular importance of the active site in Subunit C may arise from its location at the center of the assembly. Subunit C is the only subunit that is flanked on both sides by catalytically active ATPase subunits, and it makes contact with the sliding clamp at the interface between domains within a clamp subunit, rather than at the more flexible interfaces between subunits. For these reasons, ATP-driven conformational changes in subunit C may be particularly important in driving the transition from the DNA-free to the DNA-bound state.

In the simulation of the wild-type clamp loader, the interfacial residues centered on the ATP bound to subunit C exhibit lower fluctuations in structure when compared to the other subunits. For the Q118A mutant, there is increased dynamics at this active site when compared to the wild-type simulations. The absence of the glutamine sidechain at position 118 in the mutant increases the dynamics of the Asp 110 residue (*Figure 8D,E* and supplement 1). In the wild-type protein, the sidechain of Asp 110 forms a hydrogen bond with the backbone of Asn 139, the Sensor one residue that is important for catalysis, which exhibits increased dynamics when Q118 is mutated (*Figure 8F*). Mutation to Q118 also disrupts the interaction with the adjacent subunit since Asp 110 directly interacts with the interfacial residue Arg 122 on the next subunit (*Figure 8G*). We surmise that the loss of rigidity in the mutant clamp-loader is correlated with changes in the active site that may reduce the coupling between ATP binding and DNA recognition. As discussed below, the Q118N mutation reduces expression of the clamp-loader subunits and also appears to prevent the proper assembly of the complex. The molecular dynamics simulations discussed here suggest that even if the Q118N clamp loader were to assemble properly with DNA and the sliding clamp, its ability to maintain a rigid coupling between ATP-binding sites may be compromised.

## Identification of a mutation that rescues the effect of mutating Gln 118

We carried out an exhaustive search for single mutations that rescue the loss of fitness that results from mutation of Gln 118. Many of the mutations of Gln 118 result in a very substantial loss of function, with phage replication reduced to a negligible rate. We assumed that it might be difficult to identify single mutations that rescue such near-complete loss of function (the mean fitness measurement of all amino acid substitutions to Q118, measured in an independent experiment with a focused library of point mutations at position 118, is F = −1.7, *Figure 6C*). We therefore fixed a conservative substitution – Q118N – in the clamp loader sequence, with an intermediate effect on fitness, and then carried out a saturation mutagenesis screen with this background. The shorter sidechain of asparagine cannot form the hydrogen bonds with the backbone of the Arg-finger helix, but the effect of the Q118N mutation is milder (F = −1.17 ± 0.05, *Figure 6—figure supplement 1*) than other mutations at this site. We performed a second-site suppressor screen of the Q118N mutant clamp loader, screening almost all possible single mutations in the background of the Q118N mutant.

Using mutagenic primers, we introduced Q118N as the background mutation into the deep mutational scanning library of the AAA+ module of gp44. While this method excludes double mutations in a ± 5 amino acid window around position 118, it allows us to scan all other positions in the AAA+ module (220 of 230 positions). We screened the library of double mutants through three rounds of selection to amplify the gain-of-function variants. The first round of selection was started by adding T4$^{del}$ to bacteria with the mutant library. The second and third rounds of selections were performed by using the recombinant phage T4 to infect bacterial cells that do not contain the T4 replication genes. To avoid the confounding effect of random mutations that might occur during selection, we performed two independent trials of the double-mutant screen and considered only those variants that confer gain of function in both trials.

The fitness values from the two replicate trials are shown in the form of a two-dimensional histogram in *Figure 9A*. The data show that only one mutation – G143N – is able to restore a substantial

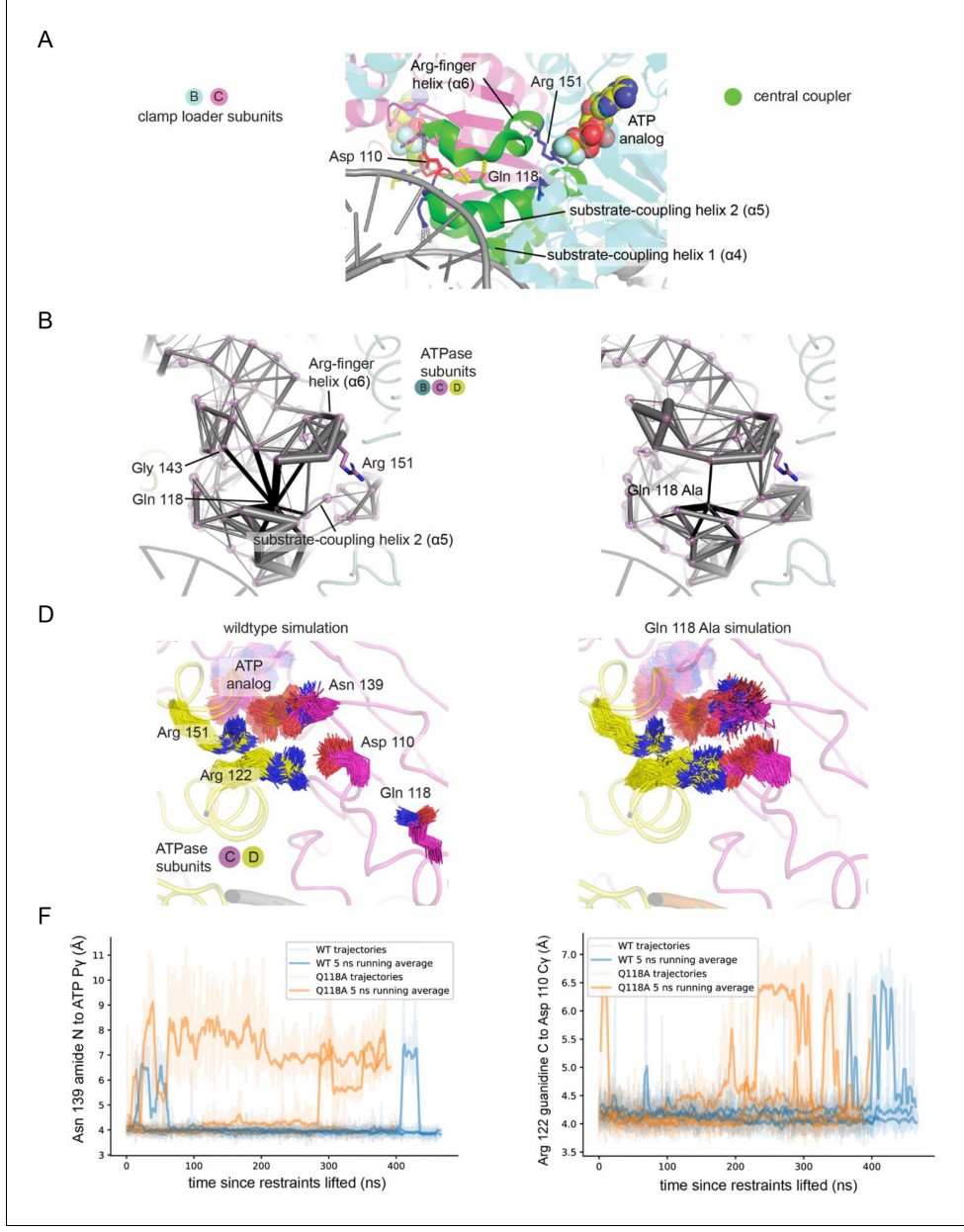

**Figure 8.** Molecular dynamics simulations of the of the wildtype T4 clamp loader and Q118A mutant. (**A**) The central-coupler unit of the clamp loader is shown in green. Pairs of residues in subunit C that show correlated motions in simulations of the wildtype structure (**B**) and Q118A structure (**C**) are connected by gray cylinders (black cylinders connect residues to position 118), with the diameter of the cylinder proportional to the magnitude of correlated motions. Values are averaged from four trajectories. The sidechain positions of important residues in subunit C, from the wildtype (**D**) and Q118A (**E**) simulations. (**F**) The distance between the amide of sensor-I residue Asn 139 and the γ-phosphate of ATP bound by subunit C from four trajectories of the wild-type clamp loader and the Q118A mutant. (**G**) The distance between the γ-carbon of Asp 110 in the walker B motif of subunit C and guanidine carbon of Arg 122 of subunit D from four trajectories of the wild-type clamp loader and the Q118A mutant.

The online version of this article includes the following figure supplement(s) for figure 8:

**Figure supplement 1.** Subunit C heavy-atom root-mean-square fluctuations from AAA+ module aligned trajectories, for wildtype and Q118A trajectories.

amount of activity (the Q118N/G143N mutant has 30% of the activity of the wild-type clamp loader). In separate head-to-head competitions, we confirmed that the Q118N/G143N variant has increased fitness compared to the Q118N mutant (*Figure 9B*). Gly 143 packs against the Asp 110 sidechain in the structure of the clamp loader (*Figure 9C*). Molecular modeling suggests that an asparagine side-chain at position 143 can form hydrogen bonds with the backbone and sidechain of Asp 110, and also with the backbone of Asn 140, which is adjacent to Sensor 1 (*Figure 9C*). It is striking that the only mutation we have found that compensates for the loss of Gln 118 results in the introduction of an asparagine sidechain that can restore hydrogen bonding between the loops leading into the arginine-finger helix (α6) and the middle helix (α5). This result suggests that the importance of Gln118 arises from its ability to couple these two helices.

Gln 118 is a completely buried residue, and buried sidechains that form backbone hydrogen bonds (as Gln 118 does) are highly conserved in proteins, and are likely to be critical for the stability of the structure (*Worth and Blundell, 2010*). Thus, mutation of Gln 118 is expected to destabilize the clamp-loader structure. To determine whether this is the case we created a clamp-loader variant in which the ATPase subunit is C-terminally tagged with the fluorescent protein mCherry, allowing the expression levels of the complex to be monitored by flow-cytometric analysis of *E. coli* cells expressing the clamp-loader subunits. This analysis shows that introduction of the Q118N mutation leads to a substantial reduction in protein expression (*Figure 6—figure supplement 2*). Note that in this experiment, the proteins are expressed using a strong promoter (the T7 RNA polymerase promoter), in order to purify proteins for biochemical experimentation. The natural T4 promoter that is used in the phage-propagation assay results in very low levels of clamp-loader expression (there is no detectable fluorescence signal when mCherry is fused to the ATPase subunit). We do not know whether the Q118N mutation also leads to expression defects under these conditions.

We purified the wild-type clamp-loader complex, the Q118N mutant complex, and the Q118N/I141L double mutant that has partial recovery of fitness in the phage propagation assay (the Q118N/G143N double mutant was not purified). Gel filtration analysis shows that the wild-type complex corresponds to a single peak eluting at ~70 ml, which we assume corresponds to correctly assembled clamp loader (*Figure 9—figure supplement 1*). For both mutants, there is a peak at ~70 ml, but also an additional peak near the void volume of the column (~45 ml). For the Q118N mutant, the fraction of the protein corresponding to the properly assembled clamp loader is small, but is substantially larger for the double mutant that has increased fitness (Q118N/I141L).

We measured the rates of DNA-stimulated ATP hydrolysis for the wild type and Q118N mutant forms of the clamp loader (*Figure 9—figure supplement 2*). Addition of the mCherry tag to the wild-type clamp loader results in a slight reduction of the ATPase activity. The mCherry-tagged Q118N mutant clamp loader has a very low rate of DNA-stimulated ATPase activity (less than 10% of wild-type activity). We also measured the extent of plasmid DNA replication by the T4 replisome, using wild-type and mutant clamp loaders (*Figure 9—figure supplement 3*). As for the ATPase assay, addition of the mCherry tag to the wild-type clamp loader results in a slight reduction of replication efficiency. Introduction of the Q118N mutation leads to severe attenuation of replication efficiency, to a level comparable to that seen in the absence of the clamp loader. These data show that purification of the Q118N mutant clamp loader leads to protein assemblies that are functionally defective.

## The glutamine-mediated helical junction is preserved in diverse AAA+ proteins

The AAA+ family can be divided into five major clades that diverged from each other before the last common ancestor of the eukaryotes and the prokaryotes (*Erzberger and Berger, 2006*; *Iyer et al., 2004*; *Leipe et al., 2002*). These major groupings include the clamp-loader clade, a clade containing proteins involved in the initiation of DNA replication (e.g. DnaA, Cdc6, ORC), the 'classical AAA' clade (containing proteins such as p97 and NSF), a clade containing proteases and disaggregases (e.g. ClpX, Lon) and a clade comprising proteins with diverse functions (e.g. MCM, NtrC). We examined available structures for members of each of these groups, and found that the glutamine-mediated junction is a feature that is present in the central-coupler units of members of at least four out of the five clades – the feature may be absent in the classical AAA clade, based on structures we have examined.

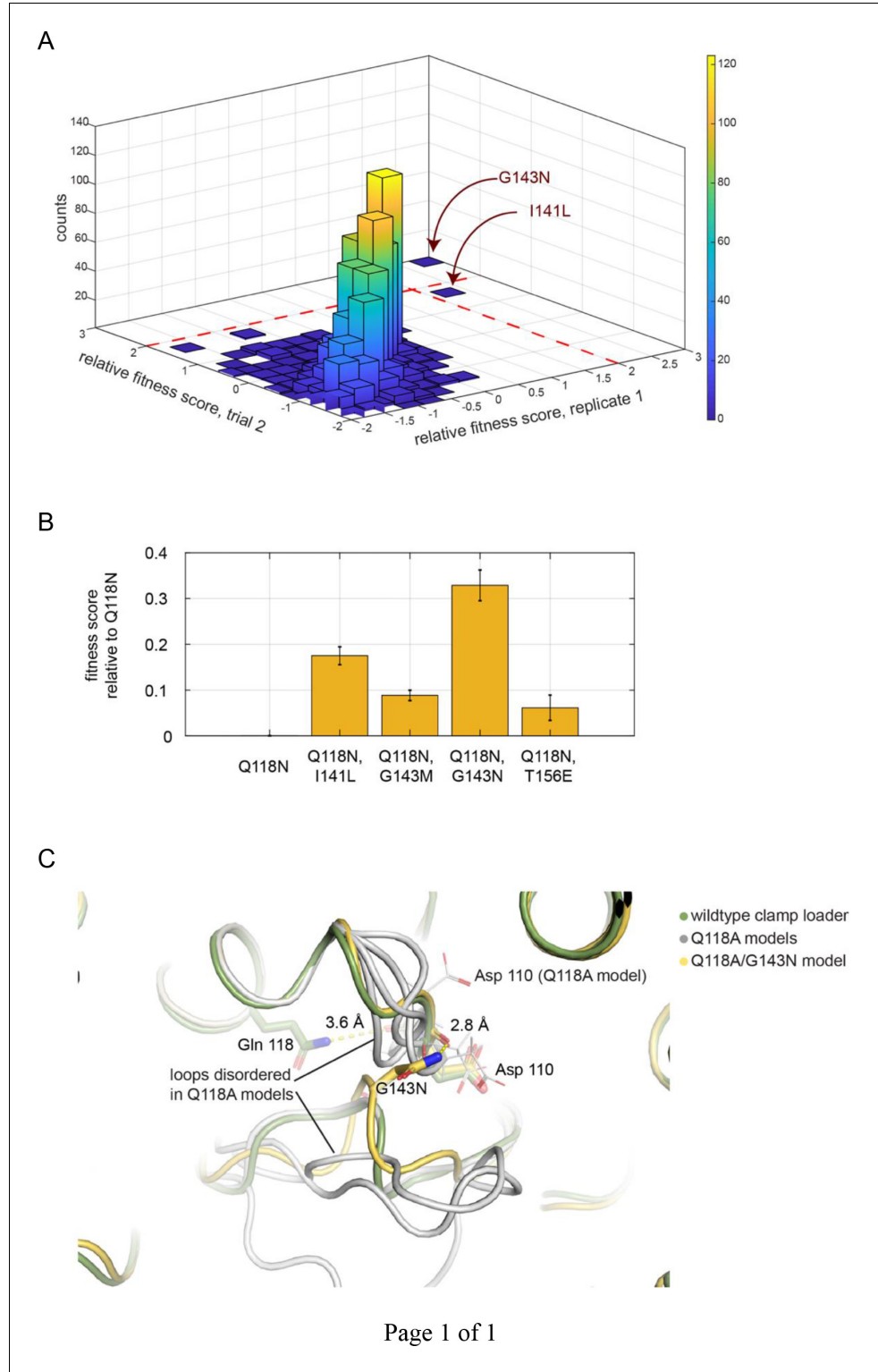

**Figure 9.** Screen for second-site suppressor mutations, in the Q118N background. (**A**) The distribution of fitness scores (relative to that for the Q118N mutant) from two independent trials. G143N is the best performing mutant in both trials. (**B**) The top four best performing mutants were tested in competition with Q118N using the phage-propagation assay. (**C**) The spatial orientation of Asp 110 is destabilized in four independent trials modeling the effect of Q118A (gray). The Q118A/G143N double mutant stabilizes and restores the orientation of Asp 110 (yellow).

*Figure 9 continued on next page*

*Figure 9 continued*

The online version of this article includes the following figure supplement(s) for figure 9:

**Figure supplement 1.** Elution profile of mCherry-tagged clamp loader variants in size-exclusion chromatography.

**Figure supplement 2.** ATPase activity of mCherry-tagged clamp loader variants, in the presence of clamp and DNA.

**Figure supplement 3.** Ability of mCherry-tagged clamp loader variants to carry out DNA replication with M13 phage ssDNA as template.

The glutamine-mediated junction is illustrated for four proteins, human RFC, ClpX, DnaA, and NtrC in *Figure 10*. In the human RFC complex, the glutamine junction is essentially the same as in the T4 clamp loader (*Figure 10A*). ClpX, the motor component of the ClpXP proteolytic complex, is particularly instructive because recent cryoEM analyses have provided structures for the hexameric ClpX protease complex bound to substrates and ATP and to the ClpP protease assembly (*Fei et al., 2020*; *Ripstein et al., 2020*). Six AAA+ modules in the ClpX complex form a spiral structure around a chain of the protein substrate that is being extruded into the ClpP protease chamber (for example, see PDB ID: 6VFS). In the ClpX assembly, the six central coupler units of the AAA+ modules form a contiguous surface around the peptide substrate, analogous to the way in which the corresponding units form a contiguous surface around DNA in the T4 clamp loader (*Figure 10E*). The hydrogen bonds made by the relevant glutamine sidechain in ClpX (Gln 208) (*Figure 10B*) bear a striking similarity with the corresponding interactions made by Gln 118 in the T4 clamp loader (*Figure 8A*). Despite differences in the details of the structure, two of these hydrogen bonds link the substrate-coupled helix 2 of the central coupler in ClpX to the Arg-finger helix, and one of them anchors the backbone carbonyl oxygen of a residue adjacent to the second aspartate of the DExD motif (Asp 187). Similarly striking correspondences are observed for DnaA (*Figure 10C*) and NtrC (*Figure 10D*), although the available structures for those proteins are incomplete in terms of either lacking the substrate in the case of NtrC, or in forming the appropriate oligomeric interactions in the case of DnaA.

For the four AAA+ clades in which members are clearly seen to have the glutamine-mediated junction, not every protein in the group has retained this feature. For example, in the clamp-loader clade, the glutamate-mediated junction is present in eukaryotic RFC proteins, which is not surprising since the T4 clamp loader is closely related to the RFC complex (*Figure 10A*). The ATPase subunit of the bacterial clamp loader, though, is lacking a residue equivalent to Gln 118, and the region of the junction between the helices is occupied by hydrophobic residues. Asp 110 in the T4 clamp loader, at the fourth position of the DExD/H motif, is replaced by histidine in the bacterial clamp loaders, and so aspects of the regulation of ATP hydrolysis and the coupling to DNA binding appear to have evolved differently. Proteins such as NSF and p97 in the classical AAA group also have a hydrophobic interface at the region corresponding to the junction, and insertions in the central-coupler unit makes it difficult to draw direct analogies to the T4 clamp loader structure. Nevertheless, our finding that the glutamine-mediated junction is preserved in at least four greatly divergent groups of AAA+ proteins establishes that this structural motif emerged very early in the evolution of the AAA+ proteins.

## Concluding remarks

The AAA+ ATPases are a remarkably diverse set of proteins, with individual members of the family engaged in widely different functions. Despite this divergence in function, the core AAA+ module is conserved, indicating that fundamental aspects of the mechanism are shared across the family. Although general aspects of the mechanisms of AAA+ ATPases are now understood, there are still considerable gaps in knowledge about how cooperativity in ATP binding and hydrolysis is converted into functionally important actions. Deep-mutagenesis methods, enabled by next-generation sequencing, can yield rigorous structure-function maps (*Shah and Kuriyan, 2019*) and provide new mechanistic insights (*Jones et al., 2020*). We therefore developed an effective and robust platform for the application of deep-mutagenesis strategies to the replication proteins of T4 bacteriophage. In this report, we focused principally on the T4 clamp-loader complex, a AAA+ ATPase that loads sliding clamps onto DNA for highly processive DNA synthesis. We analyzed the extent to which these proteins tolerate mutations while carrying out the crucial task of replicating DNA for phage

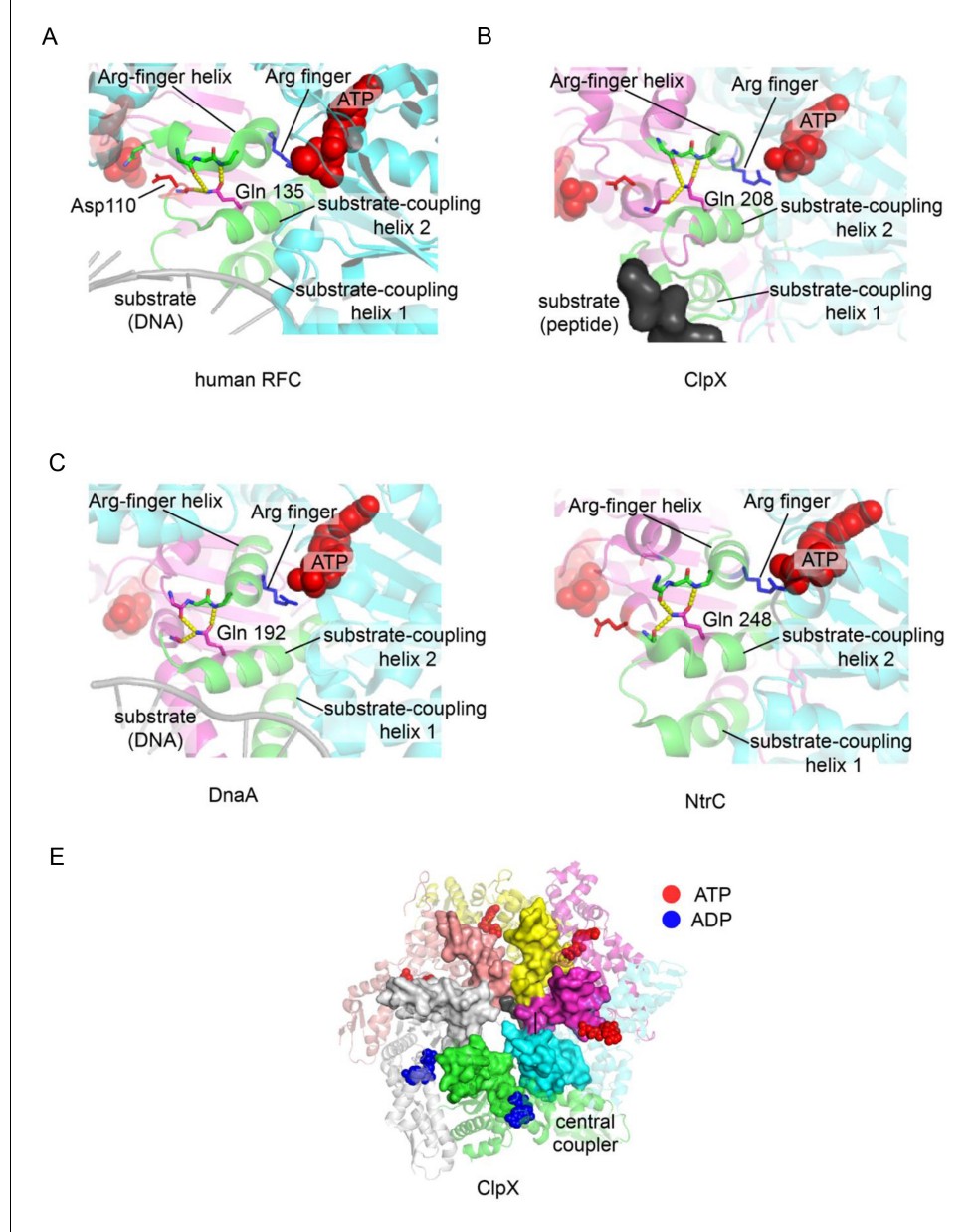

**Figure 10.** The central-coupler unit in different AAA+ proteins. The central-coupler unit of subunit C is shown in green for (**A**) the human RFC structure (PDB ID: 6vvo [*Gaubitz et al., 2020*]), (**B**) the ClpX structure (PDB ID: 6vfs [*Ripstein et al., 2020*]), (**C**) the DnaA structure (PDB ID: 3r8f *Duderstadt et al., 2011*) and of (**D**) the structure of NtrC1 (PDB ID: 3m0e [*Chen et al., 2010*]). (**E**) The central coupler in each of the subunits of ClpX form a contiguous surface lining the central channel of ClpX (PDB ID: 6vfs [*Ripstein et al., 2020*]) and contacts the peptide substrate.

propagation. These studies led us to identify a critical hydrogen-bonded network that is essential for clamp-loader function.

Our finding that the sliding clamp and the clamp loader exhibit extensive tolerance towards mutations is consistent with the results of many studies on smaller proteins (*McLaughlin Jr et al., 2012*; *Salinas and Ranganathan, 2018*; *Stiffler et al., 2015*), which document that proteins exhibit a remarkable capacity to withstand mutational insult. While residues involved in ATP coordination, ATP hydrolysis, and in DNA recognition are very sensitive to mutation, it is striking that clamp loader variants with many different point mutations distributed across other sites in the subunits can

maintain phage replication without substantial loss of function. This capacity of proteins to accommodate mutations at many sites without catastrophic loss of function has presumably enabled the rapid diversification of highly complex protein assemblies.

Comparison of the sequences of different bacteriophage clamp-loader systems shows that the residues that are mutationally tolerant in the T4 phage-replication assay are also those that accommodate sequence variation in the evolutionary history of these clamp-loader proteins. An interesting finding, which is also consistent with previous studies on smaller proteins, is that the sequence conservation data are not particularly good at predicting the effects on function of specific mutations in the T4 clamp loader. The actual mutational tolerance of the T4 proteins reflects epistatic effects that are a consequence of the particular evolutionary history and functional peculiarities of the T4 proteins. Previous work on other proteins suggests that most of these epistatic effects are likely to be local, reflecting differences in the packing of sidechains, for example. But, importantly, some also reflect alterations in allosteric mechanisms in these proteins (*Raman et al., 2016*), and they hold much potential for new structural and mechanistic insights.

Our analysis of the residues in the AAA+ module that are the most sensitive to mutation has identified the importance of Gln 118 in Domain 1 of the AAA+ module of the T4 clamp loader. Gln 118 is the only residue in the set of most-sensitive residues that is not at an ATP-binding site, is not involved in catalysis, and does not interact with DNA, the sliding clamp, or a neighboring subunit. Gln 118 is conserved in phage and archaeal clamp loaders and all the five subunits of the eukaryotic RFC complexes, but its absence in bacterial clamp loaders had led to its significance being overlooked. Gln 118 forms a hydrogen-bonded junction between the substrate-coupling helix 2 (α5) and the Arg-finger helix (α6), on the surface of Domain 1 that is distal to the ATP. These two helices, along with the substrate-coupling helix 1 (α4), form a structural unit that mediates all the important contacts between ATP, DNA, and the clamp that underlie clamp-loader function. We refer to this unit as the central coupler, given its location within the complex and its importance in coupling the effects of ATP, DNA, and clamp binding. When the clamp loader is bound to ATP, the multiple central-coupler units form a contiguous structure that encircles DNA. Rigidity in the central coupler is likely to be important for function, by allowing rapid integration of the presence or absence of ATP and DNA at each site.

The three helices that make up the central coupler are also found in the broader class of oligomeric ATPases, such as proteins of the RecA and F1-ATPase families. In those proteins too, the corresponding helices form contiguous structures that are assembled around the substrate – single-stranded DNA in the case of RecA (*Chen et al., 2008*) (see for example PDB ID: 3CMW), and the rotating central shaft of the F1-ATPase assembly (*Kagawa et al., 2004*) (see, for example, PDB ID: 1W0J). In F1-ATPase, these helices form a rigid platform that serves as a fulcrum against which the subunits are able to push while undergoing the large conformational changes that are necessary for ATP synthesis (*Böckmann and Grubmüller, 2002*; *Ma et al., 2002*). The sequences of these helices are not conserved in F1-ATPases across the span of evolution. However, there are hydrogen-bonded ionic interactions between helices that appear to play a stabilizing role. The residues that form these interactions are essentially invariant in F1-ATPases across life, suggesting that the functional rigidity is conferred mainly by the intra-helical hydrogen bonds, with critical augmentation provided by a limited set of sidechain-sidechain interactions. The residues that provide these sidechains are then extremely sensitive to mutation.

Protein structures are conserved over evolutionary time, despite sequence drift, because secondary structural elements can pack together in limited ways, and these are determined primarily by backbone geometry rather than sequence. This principle accounts for the extraordinary diversity of sequences in protein families that have conserved structures, and the AAA+ ATPases are no exception. In this work, we examined whether the functional constraints imposed by allosteric control in the clamp loader result in extensive mutational sensitivity. Our conclusion is that they do not – mutational sensitivity is restricted to a relatively small number of residues that are located primarily at the active site and interfacial regions, allowing for extensive sequence variation in the rest of the protein.

Given the crucial role of the central coupler in the clamp-loader mechanism, it is striking that the majority of residues within it are not in the set of the most mutationally-sensitive residues. Except for three structurally important glycine residues, all the residues in the most-sensitive set have sidechains that form hydrogen bonds. Of these residues, Gln 118 is the only residue with a structural

role, that of linking two of the helixes in the central coupler to each other and to the Walker B motif. The sidechain of this glutamine residue serves as a vital link that completes a hydrogen-bonded network, mainly running along the backbone hydrogen bonds of α helices, that links one ATP to the next one around the AAA+ spiral.

A reliance on α helices to maintain rigidity in the central coupler enables the AAA+ module to tolerate mutations, because the backbone hydrogen bonds of the α helices do not depend on sequence. A vulnerable point in the system appears to be the bends in the helices, or the loops leading into them, that are required to transit from the interface with the proximal ATP to the distal interface, and it is this weak point that is stabilized by the hydrogen bonds provided by Gln 118. The extraordinary diversity of AAA+ function appears to have arisen by preserving the helices of the central coupler as the architectural core responsible for cooperative ATP-driven action, while accreting additional function that is superimposed on this base. The platform we have developed makes it straightforward to perform high-throughput tests of variations in clamp-loader design. Ultimately, it is hoped that these studies will make it realistic to construct novel AAA+ proteins based on an improved understanding of how they work.

# Materials and methods

### Key resources table

| Reagent type (species) or resource | Designation | Source or reference | Identifiers | Additional information |
| --- | --- | --- | --- | --- |
| Strain, strain background (Bacteriophage T4) | T4del | This paper | | https://benchling.com/s/seq-Vg4DZh83BOrbx3RgpaaC |
| Recombinant DNA reagent | CRISPR plasmid | This paper | | https://benchling.com/s/seq-abaKV7JgTgAghRyR9ZlY |
| Recombinant DNA reagent | Donor plasmid | This paper | | https://benchling.com/s/seq-Z2Bo2vnShDbLtji83VHm |
| Recombinant DNA reagent | Helper plasmid | This paper | | https://benchling.com/s/seq-KdFkydUFMBrlRkdpORA3 |
| Recombinant DNA reagent | Recombination plasmid | This paper | | https://benchling.com/s/seq-bRQ2OUu39lSrs6I0gfmu |
| Commercial assay or kit | MiSeq 500 cycles kit | Illumina | Cat. #: MS-102–2003 | |
| Software, algorithm | FLASH | doi:10.1093/bioinformatics/btr507 | | |
| Software, algorithm | Analysis scripts | This paper | | https://github.com/kuriyan-lab/cl1 |

### Generation of T4$^{del}$ strain with deletion of clamp and clamp loader genes

The clamp gene (gene 45) and clamp-loader genes (genes 44 and 62) are located adjacent to each other on the T4 genome (32,886 bp – 30,341 bp, numbered according to the GenBank file with accession number AF158101). We could not delete these genes using CRISPR-cas9 mediated genome engineering (*Tao et al., 2017*). Our inability to edit the T4 phage genome with CRISP-cas9 is consistent with the observation that the wild-type T4 genome, with extensive hydroxymethylation and glucosylation, is recalcitrant to cas9 cleavage (*Bryson et al., 2015*; *Liu et al., 2020*; *Tao et al., 2017*; *Vlot et al., 2018*). We used CRISPR-cas12a since it was reported to efficiently cleave the genome of T4 phage in vitro (*Vlot et al., 2018*).

The published methodology to edit bacteriophage T4 (*Tao et al., 2017*) employs two plasmids – a CRISPR plasmid to cleave the genome in a targeted manner and a donor plasmid carrying the sequence to be inserted. Since T4$^{del}$ would lack essential genes, we modified the editing methodology to use an additional helper plasmid that encodes a copy of the genes to be deleted from the genome. The CRISPR plasmid, encoding the gene for CRISPR-cas12a also encodes a gRNA targeting

the clamp loader locus on the T4 genome (sequence of the target site is <u>TTTA</u>TTACTTACTTCACGA TCGAT, first four nucleotides, underlined, corresponding to the PAM sequence of CRISPR-cas12a). The donor plasmid carries the sequence to be inserted into the clamp/clamp loader locus: a new CRISPRcas12a recognition site (<u>TTTA</u>CCGGGAGGAAGATATAGCAC), surrounded by ~1 kb of the regions flanking the clamp/clamp loader locus, to act as arms of homology for recombination into the T4 genome (29,142:30,190 bp and 32922:33950 bp in the GenBank file AF158101). The helper plasmid carries a copy of the clamp and clamp-loader genes where the endogenous cas12a targeting site has been removed with synonymous mutations.

In order to generate T4$^{del}$, we infected *E. coli* bearing the three plasmids described above with wildtype T4 bacteriophage particles and poured it onto agar plates in different dilutions to be able to isolate individual plaques. We genotyped five individual plaques by PCR amplification of the clamp/clamp loader locus. Two of the plaques gave a single band on agarose gels corresponding to the deleted genome, suggesting the phage genome was edited in the early infection waves. The other three plaques were mosaic, with bands corresponding to both the deleted and wildtype genomes, suggesting that the phage genome was edited only in the later infection waves. We deemed the first two plaques to have a high fraction of deleted phage and decided to isolate a pure T4$^{del}$ phage strain from one of the plaques.

In order to isolate pure T4$^{del}$ phage, we performed another plaquing assay with *E. coli* bearing only the helper plasmid. Since there is no genome editing during plaque formation, each plaque corresponds to a 'pure' phage strain. We genotyped 10 plaques, all of which produced PCR amplicons corresponding T4$^{del}$, verified by sequencing and by a single band of the appropriate size on agarose gels. Phage from each of the ten plaques were unable to propagate in cultures of wild-type bacteria (not carrying the helper plasmid), further confirming that these phages were indeed lacking in the essential replication genes. We used phage from one of the ten plaques to generate T4$^{del}$ stocks containing $1 \times 10^9$ phages per ml (*Fortier and Moineau, 2009*).

## Construction of single-mutant libraries

We constructed comprehensive single-mutant libraries of the clamp, clasp, and the ATPase subunit of the clamp loader using oligonucleotide-directed mutagenesis of the genes encoded on the recombination plasmid. We designed two primers for each codon to be mutagenized – a sense primer and an anti-sense primer – that could be used to amplify the entire plasmid. The sense primer introduced the NNS degenerate codon at the intended mutagenic site, where N is a mixture of A, C, G, and T nucleotides and S is a mix of C and G nucleotides. Using the degenerate NNS codon in the sequence of the sense primer allowed us to use a single oligonucleotide sequence to introduce 32 of the 64 possible codons at the intended mutagenic site that codes for all twenty amino acids. The sense and antisense strands each were designed to have an overlap that corresponded to an annealing temperature between 60-65C (nearest-neighbor [*SantaLucia et al., 1996*]) and ended in G or C to promote specific and strong binding at the 3' end. The 5' ends of both primers contained the sequence GGTCTC, the recognition motif for BsaI restriction enzyme used in Golden Gate cloning (*Engler and Marillonnet, 2014*).

We ordered a total of 731 primer pairs, one per residue position of the clamp, clasp and the ATPase subunit, and performed as many PCRs for mutagenesis. All reactions yielded bands of the desired size (corresponding to linear, 5.5 kb dsDNA) on agarose gels. Most PCRs gave a single band, with intensities similar to other single bands. We pooled the PCR products together to be in equimolar ratio from each reaction, estimating concentrations by eye based on band intensities and appropriately adjusting values for the faint bands. We generated a total of 7 pools, corresponding to residue positions 2–116 and 114–228 of clamp, 2–96 and 94–186 of clasp and 2–118, 116–230 and 228–318 of the ATPase subunit. We chose the pool size to be less than 450 bases long so that each pool can be fully sequenced by the 500 cycle Illumina paired-end sequencing kit.

We gel-purified the pooled PCR products and performed a one-step restriction digestion (with BsaI) and ligation (with T4 DNA ligase) in the ligase buffer to circularize the linear PCR products. We added DpnI enzyme to the digestion-ligation mix to degrade the original replication plasmid that was used as template in the PCRs. We transformed the ligated DNA into NEB 10-beta electrocompetent *E. coli* to get >500X coverage for each pool. We isolated and saved miniprepped plasmids from each pool to be used in the phage replication assay. We sequenced each pool using the

Illumina 500 cycles kit to check library coverage and observed a few positions to be poorly represented. We repeated the mutagenesis for these positions and mixed them into the appropriate pool.

## Phage-propagation assay

The phage-propagation assay involves transforming the plasmid libraires into bacterial cells, performing the phage infection and sequencing the recombination locus on the phage genome to obtain the frequency of variants. We transformed the plasmid library of variant clamp loader or clamp genes, at >500X coverage, into electrocompetent cells of the BL21 strain of *E. coli* carrying the Crispr plasmid (used to generate T4$^{del}$ phage) that targets the cas12a site that was introduced into the genome of T4$^{del}$.

Exponentially growing cells in 100 ml of LB broth, at OD 0.1 and in a shaker maintained at 37C, were infected, in triplicate trials, with T4$^{del}$ at an MOI of 0.001. At this MOI, ~$10^6$ out of $10^9$ cells will be infected, with fewer than 500 cells getting infected by two or more phage particles. The infection was allowed to proceed for 12–16 hr, after which the infected culture flasks were allowed to sit on the bench for the cell debris to settle.

The recombination locus was prepared for sequencing through three, sequential PCR steps, each of which contains only 20 cycles of amplification to reduce biases that can be introduced by PCR. First, 5 µl of the clear supernatant from each infected flask was used as template in a 50 µl PCR to amplify only the recombined locus. One of the primers used in this step, with a sequence CTGAA TGCACACATTCGTTTGAACAGC, anneals on the genome of T4$^{del}$ outside the recombination locus. The second primer, with a sequence of CTTCAGGTTTTTTAAGAGTAATTTCAATC, anneals on the region of the plasmid that is integrated into T4$^{del}$. The pair of primers used in the first PCR ensure the amplification of variants from the library that get integrated into the phage genome, without amplification of the non-recombined plasmid molecules and the non-recombined T4$^{del}$ genome. The PCR amplicon after the first step is ~2.8 kb and includes the regions mutagenized in the seven pools. In the second PCR, we used pool-specific primers to amplify the ~350 bp corresponding to the region that is mutagenized. The primers used in the second PCR step each have 5' overhangs overlapping with Illumina adapter sequences to act as PCR handles in the next step. In the third PCR, primers introduce unique TruSeq indices and generate a ~450 bp amplicon for sequencing on the MiSeq sequencer using the 500 cycles kit.

## Protein expression and purification

### T4 clamp-loader purification

The clamp loader complex (gp44 and gp62 with a C-terminal Hisx6 tag) within a pET24 vector was co-expressed in *E. coli* BL21 cells via chemical induction with IPTG (16 hr at 18°C). Cells were resuspended in 20 mM Tris pH 7.5% and 10% glycerol and stored at −80°C until purification. For purification, cells were thawed and lysed via sonication. The soluble fraction was harvested via centrifugation at 17,000 rpm for 30 min and filtered to 0.2 micron. The supernatant was loaded onto a HisTrap column (GE Healthcare) and eluted with 500 mM imidazole. The C-terminal 6-His tag on gp62 was removed via overnight digestion with Precission Protease during dialysis into low-salt buffer (20 mM Tris pH 7.5, 10% glycerol, 150 nM NaCl and 2 mM DTT). The final protein complex was isolated via a tandem Sepharose Fast FLow and heparin column, where impurities were captured on the SP-FF column and clamp loader was eluted from the heparin column. The purified clamp loader was concentrated to ~10 mg/mL and stored in 20 mM HEPES, pH7.5, 150 mM NaCl, and 10 mM DTT at −80°C.

Mutants of gp44 were created via QuikChange mutagenesis protocols directly within the pET24 construct. No modifications of the purification protocol were necessary..

### mCherry-tagged clamp-loader purification

mCherry was cloned to be C-terminal to the ATPase subunit in the expression construct reported above, transformed into *E. coli* BL21(Rosetta-DE3) cells, and induced with 1 mM IPTG (16 hr at 18° C). Cells were resuspended in 20 mM Tris pH 7.5% and 10% glycerol, and stored at −20°C until purification. For purification, cells were thawed and lysed via sonication. The soluble fraction was harvested via centrifugation at 17,000 rpm for 30 min and filtered to 0.2 micron. The supernatant was

loaded onto a HisTrap column (GE Healthcare) and eluted with 500 mM imidazole in buffer (50 mM Tris, 200 mM NaCl, 5 mM β-mercaptoethanol, 10% glycerol at pH 7.5). The C-terminal 6-His tag on gp62 was removed via overnight digestion with Precission Protease during dialysis into low-salt buffer (20 mM Tris pH 7.5, 10% glycerol, 150 nM NaCl and 2 mM DTT). The dialyzed sample was subjected to a tandem Sepharose Fast FLow and heparin column, where impurities were captured on the SP-FF column and clamp loader was eluted from the heparin column. The final clamp loader complex was purified on the HiLoad 16/600 Superdex 200 prep grade column (GE Healthcare). The purified clamp loader was concentrated to ~10 mg/mL and stored in 20 mM HEPES, pH7.5, 150 mM NaCl, and 10 mM DTT at −80°C.

## T4 clamp purification (gp45)

The T4 clamp was expressed from a pTL151w vector within BL21 pLysS cells. Growth media was supplemented with glucose prior to expression. Protein was expressed via heat induction at 42°C for 2.5 hr. The resulting cells were resuspended in 20 mM Tris pH 7.5, 10% glycerol, and 2 mM DTT and stored at −80°C. Cells were lysed via sonication and centrifuged at 17,000 rpm for 30 min. The supernatant was separated on a Q-FF column via a 0–40% gradient of 1 M NaCl. The fractions containing gp45 were dialyzed for 3 hr into low salt buffer (20 mM Tris pH 7.5, 10% glycerol, 2 mM DTT) and passed over a Hitrap Heparin column. The flow through contained gp45 and was dialyzed overnight into buffer containing 40 mM sodium acetate pH 5.5, 10% glycerol and 2 mM DTT. The dialyzed protein was loaded onto coupled Sepharose Fast FLow, Heparin, and Q-FF columns. The protein was washed with buffer A (20 mM sodium acetate pH 5.5, 10% glycerol, and 2 mM DTT). SP-FF and heparin columns were then removed and the Q-FF column was washed with buffer B (20 mM Tris pH 8.8, 10% glycerol, and 2 mM DTT). The T4 clamp was eluted from the Q-FF column with a 0–40% gradient of buffer B to buffer B supplemented with 1 M NaCl. The final clamp protein was further purified via size exclusion through a S200/16/60 column in buffer containing 20 mM Tris pH 7.5, 150 mM NaCl, and 4 mM DTT. The purified protein was concentrated to ~20 mg/mL and stored at −80°C.

## Flow cytometric analysis of clamp-loader expression

*E. coli* BL 21 (Rosetta-DE3) cells expressing the wildtype or Q118N variant of the mCherry-fused clamp-loader complex were induced with 1 mM IPTG (16 hr at 18°C). The cultures were diluted to ~50,000 cells per ml in PBS. Approximately 150 µl of each culture was analyzed on a ThermoFisher Attune NxT Acoustic Focusing Cytometer. The distribution of mCherry fluorescence (635 nm excitation wavelength) was measured. Approximately 7000 cells were sorted for each variant.

## ATP hydrolysis assay

The ATP hydrolysis rates of various gp44/62 clamp loader complexes were measured using a coupled-enzyme assay that uses a spectrophotometer to measure the decrease in absorbance at 340 nm as NADH is converted to NAD⁺ during the regeneration of the ADP product of the ATP hydrolysis reaction. This regeneration of ADP to ATP is accomplished by pyruvate kinase conversion of phosphoenolpyruvate to pyruvate, which is converted into lactate by the enzyme lactate dehydrogenase, which concurrently converts NADH to NAD+, thus coupling ATP hydrolysis to NADH oxidation. Samples were aliquoted into a 96-well plate.

Each well's regeneration system contained 1 mM phosphoenolpyruvate, 0.39 mM NADH, and 1079 units per ml of pyruvate kinase and lactate dehydrogenase (*Duffy, 2016*). The reaction buffer per well contained 25 mM HEPES (pH 7.5), 0.5 M KOAc, and 6 mM Mg(OAc)$_2$ (*Pietroni and von Hippel, 2008*). The reaction also included 0.05 µM gp44/62, 5 µM gp45, and 2 µM P/T DNA per well. To initiate the reaction, ATP was added as a 1:1 ATP:Mg(OAc)$_2$ solution for a final concentration of 100 µM ATP per 100 µl well sample, and the assay temperature was maintained at 23°C after initiation (*Pietroni and von Hippel, 2008*). Absorbances were read every 30 s for 40 min, and the rate of ATP hydrolysis was calculated as follows:

$$ATPase\ rate\ (\mu M\ ATP\ /\ L*min*\mu M\ gp44/62) = -\frac{dA_{340}}{dt} \times 10^6 \mu M/M \times K_{path}^{-1} \times \mu M^{-1} ATPase$$

where $\frac{dA_{340}}{dt}$ is measured when the reaction is at steady state, and where $K_{path}$ is the molar absorption

coefficient for NADH for a given optical path length (here, $K_{path}$ = 6220 $M^{-1}cm^{-1}$) (*Duffy, 2016*; *Li et al., 2007*).

## M13 DNA replication assay

We followed the protocol in *Seville et al., 1996* to measure the extent of DNA replication carried out by clamp-loader variants. The assay involves the use of the dye PicoGreen to detect the amount of leading-strand DNA that is synthesized. Single-stranded M13 phage DNA is used as template, a complementary oligonucleotide, CTTCAAATATCGCGTTTTAATTCGAGCTTC, is used as primer. We performed leading-strand synthesis in 200 μl of buffer (50 mM NaCl, 10 mM Tris-HCl, 10 mM MgCl$_2$, 1 mM DTT at pH 7.9) containing 30 units/ml of T4 DNA polymerase (NEB), 100 μg/ml single stranded DNA binding protein (gp32) from T4 (NEB), 277 μg/ml T4 DNA clamp, 200 μg/ml clamp-loader complex (wildtype or mutant), 1.25 μg/ml M13 ssDNA, 200 μM dNTPs, 2 μM ATP, and 0.5 μM primer. After allowing the reaction to proceed for 10 min, the reaction was stopped by adding 40 μl of the dye solution (1:100 dilution of PicoGreenTM (ThermoFisher) in TE buffer). Control reactions for 100% leading-strand synthesis was setup using the Phusion polymerase system, M13 ssDNA template and the primer, with 5 min of extension at 72°C before stopping the reaction with the dye solution. Dye fluorescence was measured at 485 nm excitation/528 nm emission using a Synergy H4 Hybrid Multi-Mode Microplate Reader.

## Molecular dynamics simulations

We selected a crystal structure representing the pre-hydrolysis state of the T4 loader-clamp-DNA complex (PDB ID 3U60, [*Kelch et al., 2011*]) to simulate the state expected to have the strongest interactions between the loader and its substrates. The deposited model was refined using brief molecular dynamics flexible fitting within the ISOLDE software (*Croll, 2018*). This step resolved clashes (including close contacts within nucleotides), remodeled the T57 rotamer to coordinate the active site magnesium, modeled two missing sidechains, and modeled the four to six N-terminal residues previously unmodeled from each ATPase subunit. To model a missing flexible loop within subunit E, 1584 models of residues 225–233 were generated using the next-generation kinematic closure method within Rosetta's loopmodel application (*Stein and Kortemme, 2013*), then the four lowest-energy models were each selected as the initial model for a molecular dynamics trajectory. AM1-BCC partial charges for the particular nucleotide conformations in 3U60 were generated as parameters for Rosetta using Antechamber from AmberTools19 (*Case et al., 2019*). Naive models of the Q118A mutant were generated by simply deleting all atoms beyond the beta carbon of Q118 in all four gp44 subunits within these same models. Histidine protonation states were estimated using the H++ web server (*Anandakrishnan et al., 2012*).

These four models were each solvated with TIP3P water within a truncated octahedron 15 Å larger than the complex and neutralized with Na$^+$ using tleap (*Case et al., 2019*), then 150 mM NaCl was added based on this initial volume. The complex was parameterized using the ff14SB protein force field (*Maier et al., 2015*), parmbsc1 DNA force field (*Ivani et al., 2016*), and polyphosphate parameters (*Meagher et al., 2003*). A 9.25 Å cutoff was used for nonbonded interactions, particle mesh ewald applied for long-range electrostatics, and the SHAKE algorithm with 1e-6 Å tolerance applied to constrain bonds to hydrogen during all dynamics. The system was first minimized with 10 kcal/mol/Å2 restraints to initial complex positions within Amber18 pmemd running on CPUs (*Case et al., 2019*). 1ns of Langevin dynamics (1/ps friction coefficient) with a one fs time step was used to equilibrate the system in NVT at 310.15 K while retaining restraints, within pmemd on GPUs. Then, a Monte Carlo barostat was used with two fs time step to equilibrate in NPT with restraints for 1ns.

To ensure complete solvation of the active site and nucleotide prior to unrestrained dynamics, a Monte Carlo procedure for exchanging water molecules between bulk and sites within the protein was applied (*Ben-Shalom et al., 2019*). Waters were allowed to exchange anywhere within the simulation box, retaining restraints on the complex for 1000 NVT molecular dynamics steps between each of 5000 Monte Carlo attempts. The resulting system was again subjected to restrained minimization, followed by unrestrained minimization, and 1ns NVT equilibration with the complex restrained to these new positions. The first 10ns of unrestrained NPT following this was discarded as equilibration time for analysis, but is included in trajectories. The first 400ns beyond this equilibration

period was analyzed for each of the four wild-type trajectories, and 320ns for each of the four Q118A trajectories.

To minimize spurious (and moment-dependent) correlations arising from rigid body motions of each domain with respect to the complex's center of mass, we used alignments of only the relevant domains to estimate correlated motions between residues. Each gp44 subunit's AAA+ module (residues 1–229) was aligned across all wild-type or Q118A trajectories using CPPTRAJ (*Roe and Cheatham, 2013*), then center-of-mass coordinates for each residue (including side chain and backbone atoms) were calculated using MDTraj (*McGibbon et al., 2015*). Linear correlations between these center-of-mass coordinates were estimated within CPPTRAJ, while generalized correlations were estimated using the method and code of *Lange and Grubmüller, 2006* (linked against GROMACS 3.3). Contacts throughout each set of trajectories were determined using GetContacts, including water-mediated contacts (*Venkatakrishnan et al., 2019*). The contact_network_pymol_viz.py script from GetContacts was adapted to draw cylinders between all pairs of residues in contact in 35% of frames from all four trajectories if their correlation coefficient is greater than 0.35 for linear correlations or 0.5 for generalized correlations (reflecting different distributions of correlation values) within PyMOL (*Schrödinger, 2015*). Edges between adjacent residues were not drawn.

Both from visual inspection when displayed on the structure, and from a scatterplot of linear versus generalized correlations over all residue pairs in contact, we concluded that they contain similar information. Only linear correlations are drawn in *Figure 8*.

For determining between-subunit correlations, we additionally aligned pairs of adjacent AAA+ modules and calculated correlations between pairs of residues as before. We then augmented the previously-block-diagonal within-subunit correlation matrices with these between-subunit correlation coefficients.

## Receiver-operator curve calculations for using sequence conservation to predict the average mutational sensitivity of positions in the AAA+ module

Positions in the AAA+ module are assigned a binarized-conservation score (1 if their conservation, in bits (*Crooks et al., 2004* ) is greater than or equal to a threshold value $t$, 0 otherwise). A varying cut-off value of relative fitness score is used to binarize the average effect of mutating each position of the AAA+ module, assigning the positions as mutationally sensitive (or insensitive). For each value of the cut-off, the true and false positive rates are calculated and plotted against each other in the form of a receiver-operator characteristic curve. The true positive rate is defined as the fraction of mutationally sensitive positions (at a given cut-off value of relative fitness score) that are conserved (binarized conservation score of 1). The false positive rate is defined as the fraction of mutationally insensitive positions (at the given cut-off value of relative fitness score) that are conserved. The accuracy of using binarized-conservation scores to predict the mean-mutational sensitivity of positions is given by the area under the receiver-operator characteristic curve.

## Acknowledgements

We thank members of the Kuriyan lab for helpful discussions, and Dr. Christine Gee for help in setting up the molecular simulations. We thank Dr. Serena Muratcioglu for help in setting up flow-cytometric analysis of cells overexpressing clamp loader variants. We thank Dr. Marjorie Russel and Dr. Lucas Harrington for helpful discussions in developing the phage-propagation assay. We thank the Doudna lab for sharing CRISPR plasmids and the wildtype strain of bacteriophage T4. We thank Prof. Rama Ranganathan for insightful discussions. We thank Prof. Yamuna Krishnan for comments on the manuscript. This research used the Savio computational cluster resource provided by the Berkeley Research Computing program at the University of California, Berkeley (supported by the UC Berkeley Chancellor, Vice Chancellor for Research, and Chief Information Officer). MKO thanks the Amgen Scholars program for funding support. PR thanks the SURF L and S fellowship for funding support. KM was supported by the NIH T32 training grant (project # 5T32AI100829-07).

## Additional information

### Competing interests
John Kuriyan: Senior editor, *eLife*. The other authors declare that no competing interests exist.

### Funding

| Funder | Grant reference number | Author |
|---|---|---|
| Howard Hughes Medical Institute | | Subu Subramanian<br>Kent Gorday<br>Peter Ren<br>Xiao Ran Luo<br>John Kuriyan<br>Michael E O'Donnell |
| Amgen Foundation | | Matthew R Orellana |
| National Institutes of Health | 5T32AI100829-07 | Kendra Marcus |

The funders had no role in study design, data collection and interpretation, or the decision to submit the work for publication.

### Author contributions
Subu Subramanian, Conceptualization, Data curation, Software, Formal analysis, Validation, Investigation, Visualization, Methodology, Writing - original draft, Writing - review and editing; Kent Gorday, Data curation, Formal analysis, Visualization, Methodology, Writing - original draft; Kendra Marcus, Data curation, Methodology, Writing - original draft; Matthew R Orellana, Methodology; Peter Ren, Data curation, Methodology; Xiao Ran Luo, Data curation; Michael E O'Donnell, Conceptualization, Writing - review and editing; John Kuriyan, Conceptualization, Resources, Supervision, Funding acquisition, Writing - original draft, Project administration, Writing - review and editing

### Author ORCIDs
Subu Subramanian (iD) https://orcid.org/0000-0001-6095-7021
John Kuriyan (iD) https://orcid.org/0000-0002-4414-5477

### Decision letter and Author response
Decision letter https://doi.org/10.7554/eLife.66181.sa1
Author response https://doi.org/10.7554/eLife.66181.sa2

## Additional files

### Supplementary files
• Source data 1. Relative enrichment values of single mutants of the ATPase subunit of the clamp loader.

• Source data 2. Relative enrichment values of single mutants of the sliding clamp.

• Source data 3. Relative enrichment values of single mutants of the clasp subunit of the clamp loader.

• Transparent reporting form

### Data availability
All data generated or analyzed during this study are included in the manuscript and supporting files. Source data files with raw counts and relative fitness scores are included. Processed data and analysis scripts are available on https://github.com/kuriyan-lab/cl1.

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
