## [Decision Letter]

**Acceptance summary:**

This paper explores relationships between protein structure, function, and sequence in the bacteriophage T4 clamp-clamp loader complex, a highly studied member of the large family of AAA+ ATPase molecular machines. In general, the experimental design is excellent. The work reveals high tolerance to substitution and poor correlation between permitted substitutions and phylogenetic variation. A hitherto unrecognized residue is found to be critical for maintenance of functional structure/rigidity.

**Decision letter after peer review:**

Thank you for submitting your article "Allosteric communication in DNA polymerase clamp loaders relies on a critical hydrogen-bonded junction" for consideration by *eLife*. Your article has been reviewed by two peer reviewers, and the evaluation has been overseen by a Reviewing Editor and Cynthia Wolberger as the Senior Editor. The following individual involved in review of your submission has agreed to reveal their identity: Christopher P Hill (Reviewer #1).

Essential revisions:

One major concern is that the levels of protein expression and folding are not verified. This is concerning for the Gln118 mutation because lack of fitness could result trivially from misfolding or accelerated degradation that might result from increased flexibility and conformational stability. Moreover, the authors' finding that it was not possible to purify Gln118 mutant proteins for biochemical studies is consistent with this sort of trivial explanation for apparent lack of biological function.

---

## [Author Response]

The reviewers raised an important issue, which is that we had not analyzed the effect of a key mutation on the expression and stability of the clamp-loader complex. Our high-throughput mutagenesis of the clamp loader system had identified a mutation in the clamp loader ATPase subunit, Q118N, that results in a substantial reduction in fitness in the phage propagation assay. The importance of Gln 118 as a critical residue for clamp loader function had not been appreciated earlier. We had provided no data concerning the effects of this mutation on expression levels of the clamp loader complex, although we had noted that the mutant complex was difficult to purify, suggesting that stability was affected.

As Reviewer #1 notes:

“One major concern is that the levels of protein expression and folding are not verified. This is concerning for the Gln118 mutation because lack of fitness could result trivially from misfolding or accelerated degradation that might result from increased flexibility and conformational stability. Moreover, the authors' finding that it was not possible to purify Gln118 mutant proteins for biochemical studies is consistent with this sort of trivial explanation for apparent lack of biological function.”

As described in the manuscript, the sidechain of Gln 118 makes hydrogen bonds with the backbone segment leading into an adjacent helix. We had omitted to point out in the original manuscript that Gln 118 is completely buried in the structure (we now do so in the revised manuscript, on page 29). As shown by Worth and Blundell, buried polar sidechains that form backbone hydrogen bonds (as Gln 118 does) are highly conserved in proteins, and these polar sidechains are important for the stabilization of the protein architecture (Worth and Blundell BMC Evolutionary Biology 2010, 10:161 http://www.biomedcentral.com/1471-2148/10/161). Thus, we do expect the mutation of Gln 118 to destabilize the clamp loader structure. However, we do not find the identification of the importance of Gln 118 to be a *trivial* finding, because the role of polar residues in maintaining structure is quite commonly linked to their functional role, making it difficult to separate the two effects. For example, the proximal histidine that links the F helix in hemoglobin to the iron atom is perhaps the most important residue for allosteric communication in hemoglobin. Mutation of the proximal histidine severely destabilizes hemoglobin, due to loss of heme binding and conversion to a molten globule state (see, for example, Brennan and Matthews, *Hemoglobin*, 21:393-403, 1997; https://doi.org/10.3109/03630269708993126).

It was an oversight for us to have not analyzed the effects of Q118 mutations on stability and function, and we have now rectified this. We now include the results of the following four experiments, in which we compare the expression of the mutant and wild-type forms of the clamp loader, their behavior on gel filtration analysis, and their activities in ATPase assays and DNA replication assays. These experiments demonstrate that the mutation most likely destabilizes the protein, and affects the nature of the assembled complex. These results further emphasize the crucial nature of the hydrogen-bonding interactions made by the Gln 118 sidechain.

1. We created a clamp loader variant in which the ATPase subunit is C-terminally tagged with the fluorescent protein mCherry, allowing the expression levels of the proteins to be monitored by flow cytometry of *E. coli* cells. This experiment shows that introduction of the Q118N mutation leads to a very substantial reduction in protein expression (Figure 6 supplement 2 in the revised manuscript).

An important point is that the proteins are expressed using a strong promoter (T7 RNA polymerase promoter), which was done so as to purify proteins for biochemical experimentation and also enable ready detection of the mCherry fluorescence. The natural T4 promoter that is used in the phage assay results in very low levels of protein expression (no detectable fluorescence signal when mCherry is fused to the ATPase subunit), and we do not know whether the expression defect that we see is also manifested under conditions where the protein expression is low. Nevertheless, the data do indicate that the Q118N mutation destabilizes the clamp loader complex.

2. We purified mCherry tagged variants of the wild-type clamp-loader complex, the Q118N mutant complex, and the Q118N/I141L double mutant that has partial recovery of fitness in the phage propagation assay. SDS-PAGE analysis (not shown) confirms that all complexes have the ATPase and clasp subunits of the clamp loader in the proper 4:1 ratio. Gel filtration analysis shows that the wild-type complex corresponds to a single peak eluting at ~70 ml, which we assume corresponds to correctly assembled clamp loader (see Figure 9 supplement 1 in the revised manuscript). For both mutants, there is a peak at ~70 ml, corresponding to the properly assembled clamp loader, but also an additional peak that is close to the void volume of the column (~45 ml). For the Q118N mutant, the fraction of the protein corresponding to the properly assembled clamp loader is small. This fraction is substantially larger for the double mutant that has increased fitness (Q118N/I141L), indicating that one effect of the second mutation is to recover the ability of the clamp loader to assemble properly.

3. We measured the rates of DNA-stimulated ATP hydrolysis for purified and mCherry-tagged wild-type clamp loader and the Q118N mutant, as we had described in the original manuscript for several other mutants (Figure 9 supplement 2 in the revised manuscript). Addition of the mCherry tag to the wild-type clamp loader results in a slight reduction of the ATPase activity. The Q118N mutation has a very low rate of DNA-stimulated ATPase activity (less than 10% of activity of the wild-type mCherry-tagged clamp loader). These data indicate that even in the fraction of Q118N mutant that can be purified as part of an intact clamp loader complex, the mutation compromises the ability to hydrolyze ATP. This is likely to be due to the failure to assemble into a competent conformation.

4. We measured the extent of plasmid DNA replication by the T4 replisome, using wild-type and mutant clamp loaders, as described in the original manuscript (Figure 9 supplement 3 in the revised manuscript). As for the ATPase assay, addition of the mCherry tag to the wild-type clamp loader results in a slight reduction of replication efficiency. Introduction of the Q118N mutation leads to a near-total loss of replication efficiency, to a level comparable to that seen in the absence of the clamp loader.

The main text of the manuscript now includes a description of these new results, and the new data are included as supplementary figures.